# LIFe-GoM: Generalizable Human Rendering with Learned Iterative Feedback Over Multi-Resolution Gaussians-on-Mesh

**Jing Wen, Alexander G. Schwing & Shenlong Wang**
University of Illinois Urbana-Champaign
{jw116, aschwing, shenlong}@illinois.edu
**https://wenj.github.io/LIFe-GoM/**

## Abstract

Generalizable rendering of an animatable human avatar from sparse inputs relies on data priors and inductive biases extracted from training on large data to avoid scene-specific optimization and to enable fast reconstruction. This raises two main challenges: First, unlike iterative gradient-based adjustment in scene-specific optimization, generalizable methods must reconstruct the human shape representation in a single pass at inference time. Second, rendering is preferably computationally efficient yet of high resolution. To address both challenges we augment the recently proposed dual shape representation, which combines the benefits of a mesh and Gaussian points, in two ways. To improve reconstruction, we propose an iterative feedback update framework, which successively improves the canonical human shape representation during reconstruction. To achieve computationally efficient yet high-resolution rendering, we study a coupled-multi-resolution Gaussians-on-Mesh representation. We evaluate the proposed approach on the challenging THuman2.0, XHuman and AIST++ data. Our approach reconstructs an animatable representation from sparse inputs in less than 1s, renders views with 95.1FPS at $1024 \times 1024$, and achieves PSNR/LPIPS*/FID of 24.65/110.82/51.27 on THuman2.0, outperforming the state-of-the-art in rendering quality.

## 1 Introduction

Generalizable rendering of an animatable human avatar from sparse inputs, i.e., images showing a human in the same clothing and environment but not necessarily the same pose, is an important problem for augmented and virtual reality applications. Envision generation of an animatable avatar from a few quickly taken pictures in an unconstrained environment and efficient yet high-quality pose-conditioned rendering in a virtual world.

To address this application, recent approaches (Kwon et al., 2024; Zheng et al., 2024; Li et al., 2024; Hu et al., 2023; Pan et al., 2023) resort to generalizable reconstruction methods. Generalizable methods avoid scene-specific optimization at inference time but instead use *just* a single deep net forward pass, making reconstruction efficient. During an offline training phase the deep net extracts data priors and inductive biases from a reasonably large dataset. Due to the learned priors, it can be applied to sparser inputs compared to scene-specific training.

For rendering, recent methods (Wen et al., 2024; Paudel et al., 2024; Guédon & Lepetit, 2024) introduce a dual shape representation, combining the advantages of a mesh, i.e., regularization via the manifold neighborhood connectivity induced by the triangle mesh, with those of Gaussian splats, i.e., fast and flexible rendering.

However, the use of *just* a single deep net forward pass during reconstruction prevents present-day methods from refining their prediction. This is a concern because apparent errors that can be detected by comparing available inputs to a corresponding rendering of the reconstruction are not utilized. Moreover, w.r.t. the dual shape representation for human rendering, GoMAvatar (Wen et al., 2024) and iHuman (Paudel et al., 2024) employ identical resolutions for the underlying mesh and Gaussians, i.e., one Gaussian for each triangle face in the mesh. This is a concern because a

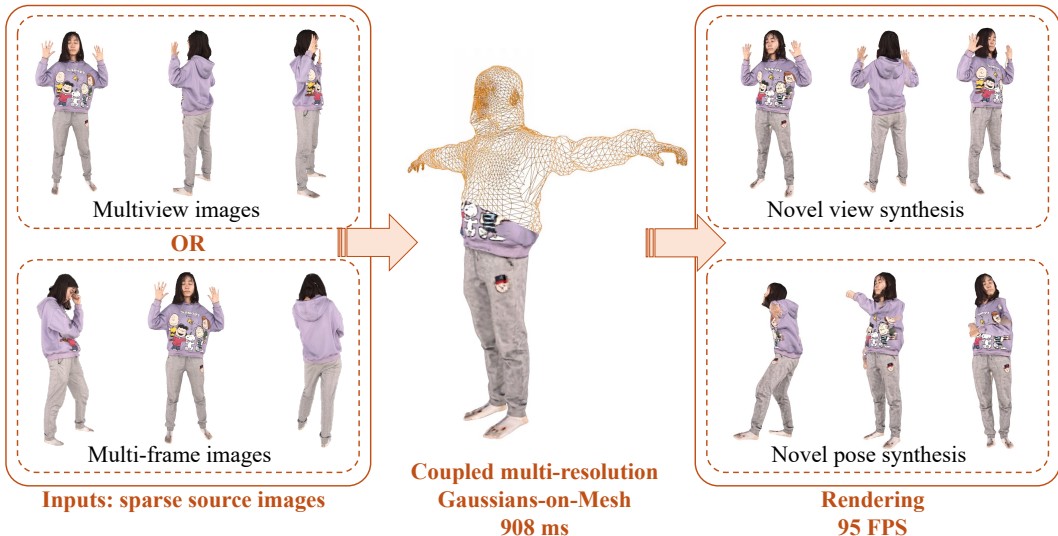

Figure 1: **Overview.** We tackle the problem of generalizable human rendering. Given sparse source images (multiview images or multi-frame images), we reconstruct the 3D human representation in canonical T-pose space. The canonical representation can be animated and rendered in novel views.

reasonably low-dimensional mesh representation is desirable for efficient reconstruction, while a high-dimensional Gaussian splat representation is desirable for high-quality rendering. GaussianAvatar (Qian et al., 2024a) uses an adaptive density control based on gradients to densify Gaussians on the mesh. However, generalizable human rendering reconstructs and renders subjects in a feed-forward pass and therefore gradients are unavailable to guide the densification.

To address the first concern of not leveraging apparent errors, we propose a novel iterative feedback-based reconstruction network. The iterative update mechanism augments generalized methods via a feedback mechanism to improve results by fusing information from inputs, the current 3D reconstruction, and current rendering from input views. Importantly, the designed iterative update mechanism is end-to-end trainable, i.e., the feedback is taken into account when training the generalized reconstruction. Note that the iterative update mechanism makes reconstruction slightly slower, yet our un-optimized version still performs the task in less than one second. Since reconstruction is a one-off task, independent from pose-conditioned rendering, we think it makes sense to spend a bit more effort than a simple deep network.

To address the second concern, we study a coupled-multi-resolution Gaussians-on-Mesh representation. More specifically, reconstruction is performed with a low-resolution mesh while we increase the number of Gaussians by attaching multiple ones to a single triangle face. This is achieved via a sub-division-like procedure. Beneficially, reconstruction remains efficient while rendering can achieve high-quality and high-resolution results.

We illustrate our method in Fig. 1 and observe compelling rendering quality and speed. We assess the efficacy of the proposed method on the challenging THuman2.0, XHuman and AIST++ data. As mentioned, reconstruction needs less than one second and rendering runs at 95.1 FPS on one NVIDIA A100 GPU. The rendering quality of the designed method outperforms the state-of-the-art, improving PSNR/LPIPS*/FID to 24.65/110.82/51.27 from 21.90/133.41/61.67 for GHG (Kwon et al., 2024).

## 2 RELATED WORK

Rendering of human avatars can be broadly categorized into two main areas: 'per-scene optimized human rendering' and 'generalizable human rendering'. We review both areas next before discussing prior work on dual shape representations, which combine Gaussians and meshes, and iterative networks.

**Per-scene optimized human rendering.** Human rendering from multiview or monocular videos has achieved great results in recent years, benefitting from progress in neural rendering, e.g., neural radiance fields (NeRF) (Mildenhall et al., 2020) and Gaussian splatting (Kerbl et al., 2023).

NeuralBody (Peng et al., 2021) is one of the earlier works that explores NeRFs for human rendering. It regresses the colors and opacities based on the latent codes associated with the vertices of a deformable mesh. HumanNeRF (Weng et al., 2022) learns subject-specific representations from a monocular video and improves over prior works by introducing non-rigid transformations. Followup NeRF-based works further improve the rendering quality (Yu et al., 2023), training speed (Geng et al., 2023; Jiang et al., 2023), and rendering speed (Jiang et al., 2023). Later, Gaussian splatting was adopted by human rendering techniques due to its superior rendering speed (Lei et al., 2024; Wen et al., 2024; Hu et al., 2024; Kocabas et al., 2024; Li et al., 2023; Paudel et al., 2024). Human101 (Li et al., 2023) advances the training speed to ∼100s on ZJU-MoCap and MonoCap. iHuman (Paudel et al., 2024) further improves the training speed to 12s on PeopleSnapshot and can be trained on as few as 6 frames. Even though the training speed improves significantly when using Gaussian splatting, real-world applications often prefer sub-second training times. Moreover, without learned priors from large-scale datasets, per-scene optimization approaches suffer from overfitting when the training views are sparse.

Differently, in this work, we adopt the dual shape representation introduced by GoMAvatar (Wen et al., 2024) and adapt it to generalizable human rendering. This permits to reconstruct the 3D representation in less than one second and further excels even if only sparse inputs are available.

**Generalizable human rendering.** Generalizable human rendering operates on sparse source views and benefits from learned priors and inductive biases extracted during a training phase from large-scale datasets. In addition, it has a greater potential to attain a faster speed when recovering a 3D representation from the source views. ActorsNeRF (Mu et al., 2023) combines per-scene optimization with priors learned from large-scale datasets using a two-stage training. Diffusion-based approaches and large-reconstruction model-based methods (Weng et al., 2024; Chen et al., 2024; Xue et al., 2024; Kolotouros et al., 2024; Pan et al., 2024) denoise the multiview images or other properties. Since it requires multiple steps for each denoising process, diffusion-based approaches usually take 2-10s to reconstruct the human avatar from images. Another line of works (Remelli et al., 2022; Hu et al., 2023; Kwon et al., 2021; 2023; Li et al., 2024; Pan et al., 2023; Zheng et al., 2024) build a single feed-forward approach to recover a 3D representation. They operate on source views and output a 3D representation for novel view rendering. Without evaluating the network several times, feed-forward methods are much faster compared to diffusion-based methods.

Our approach falls in the feed-forward category. However, differently, we devise an end-to-end trainable iterative feedback module to improve performance. As we show quantitatively and qualitatively in Section 4, our approach achieves better rendering quality compared to prior feed-forward methods, while not being significantly slower.

**Gaussians-on-Mesh dual shape representation.** Though Gaussian splatting alone achieves superior rendering quality and speed, it suffers from overfitting when a good position initialization is not available (Wen et al., 2024) and its underlying geometry is less accurate (Paudel et al., 2024; Qian et al., 2024a). Prior work (Wen et al., 2024; Paudel et al., 2024) regularizes the Gaussians and enables animation using parametric models such as FLAME (Li et al., 2017) and SMPL (Loper et al., 2015). We also combine Gaussian splatting with a mesh. Different from the use of one Gaussian per face by Wen et al. (2024) and Paudel et al. (2024), we adopt a coupled-multi-resolution representation: a low-resolution mesh is deformed and Gaussians are linked to a high-resolution mesh. Different from Qian et al. (2024a), who split the Gaussians based on gradient signals, we subdivide the mesh and bind the Gaussians on the subdivided mesh since gradients are unavailable in our generalized human rendering setting which uses only a feed-forward pass. SuGaR (Guédon & Lepetit, 2024) works on general *static* scenes and attaches multiple Gaussians to each triangle based on predefined barycentric coordinates. However, the Gaussians' scales are learned in the world coordinates, while we define Gaussian parameters in a triangle's *local* coordinates. This modification is important for modeling dynamic scenes.

**Iterative network.** Our approach falls into the category of iterative feedback networks (Adler & Öktem, 2017; Manhardt et al., 2018; Carreira et al., 2016; Li et al., 2018; Ma et al., 2020). The core idea is to learn to iteratively update the output through a forward process. This method works

particularly well when feedback signals can be incorporated at each step to improve the estimation. Previous works either unrolled standard optimizers into differentiable feedforward networks (Wang et al., 2016; Belanger & McCallum, 2016; Schwing & Urtasun, 2015; Zuo & Deng, 2025), explicitly optimizing an energy function, or trained a generic iterative network with supervised learning without an explicit energy formulation (Andrychowicz et al., 2016; Wichrowska et al., 2017; Flynn et al., 2019; Teed & Deng, 2020). In computer vision, these methods have been used for pose estimation (Li et al., 2018; Carreira et al., 2016), inverse problems (Ma et al., 2020), dense reconstruction (Flynn et al., 2019), optical flow (Teed & Deng, 2020), and depth estimation (Zuo & Deng, 2025). Our work presents a novel use of this iterative framework for generalizing human avatars.

## 3 METHOD

In the following we first provide an overview of the proposed approach in Section 3.1. We then detail our two contributions: first the coupled-multi-resolution Gaussians-on-Mesh representation in Section 3.2 and then our reconstruction approach with iterative feedback in Section 3.3. Finally we provide some information on training of the proposed method in Section 3.4.

### 3.1 OVERVIEW

**Input.** The proposed method operates on a set of source images $\{I_n\}_{n=1}^N$, corresponding binary source masks $\{M_n\}_{n=1}^N$ identifying the human, source camera extrinsics $\{E_n\}_{n=1}^N$, source camera intrinsics $\{K_n\}_{n=1}^N$, and human poses $\{P_n\}_{n=1}^N$. Here, $N$ is the number of source images. The human pose $P_n = (R_n^j, T_n^j)_{j=1}^J$ is represented by a collection of $J$ rotations $R_n^j$ and translations $T_n^j$.

**Output.** Given this input, our goal is to render the target image $I_{\text{tg}}^{\text{pred}}$ and its corresponding binary mask $M_{\text{tg}}^{\text{pred}}$ given as additional input the target camera extrinsics $E_{\text{tg}}$, intrinsics $K_{\text{tg}}$, and the target human pose $P_{\text{tg}}$, again specified via a collection of $J$ rotation matrices and translation vectors.

**Method overview.** We render $I_{\text{tg}}^{\text{pred}}$ and $M_{\text{tg}}^{\text{pred}}$ by transforming a learned canonical Gaussian-on-Mesh representation $\text{GoM}^c$ specified in a T-pose space. For this, Gaussians and mesh (i.e., $\text{GoM}^c$) are first articulated using the target pose $P_{\text{tg}}$ and subsequently transformed to target image space via the target camera parameters. We provide details in Section 3.2 and formally write this as

$$I_{\text{tg}}^{\text{pred}}, M_{\text{tg}}^{\text{pred}} = \texttt{Renderer}(\text{GoM}^c, P_{\text{tg}}, E_{\text{tg}}, K_{\text{tg}}). \tag{1}$$

The canonical 3D representation $\text{GoM}^c$ is extracted from the $N$ source images. We abstract this via

$$\text{GoM}^c = \texttt{Reconstructor}(\{I_n\}_{n=1}^N, \{M_n\}_{n=1}^N, \{P_n\}_{n=1}^N, \{E_n\}_{n=1}^N, \{K_n\}_{n=1}^N), \tag{2}$$

and provide details in Section 3.3. Unlike GPS-Gaussian (Zheng et al., 2024), we choose to reconstruct the subject in the canonical T-pose instead of the poses provided as an input. Benefitting from this choice, our representation can be retargeted to novel poses without any post-processing, such as skeleton binding. Further, our model can operate on images showing different poses. Notably, our $\text{GoM}^c$ representation uses different resolutions for the Gaussians and the mesh, and the $\texttt{Reconstructor}$ benefits from an iterative feedback update.

### 3.2 COUPLED-MULTI-RESOLUTION GAUSSIANS-ON-MESH REPRESENTATION

In this section, we describe the details of the $\texttt{Renderer}$ used in Eq. (1). We first define the coupled-multi-resolution Gaussians-on-Mesh representation in Section 3.2.1, which refers to our canonical T-pose shape. Next, we detail articulation and rendering in Section 3.2.2 and Section 3.2.3.

### 3.2.1 CANONICAL REPRESENTATION

The classic Gaussians-on-Mesh (GoM) representation associates one Gaussian with one triangle face of a mesh, i.e., the number of Gaussians is identical to the number of triangle faces. Further note, in GoMAvatar (Wen et al., 2024), the vertices of the mesh and the Gaussians' parameters in the triangle's *local* coordinates are optimized per scene.

To achieve high-quality rendering, GoMAvatar subdivides the mesh to increase the number of Gaussians. However, in the generalizable human rendering setting, naively subdividing the mesh significantly increases the reconstruction time from less than 1s to ∼13s since the network operates on a larger set of points. We therefore study the coupled-multiresolution Gaussians-on-Mesh representation. It reduces the computational cost while simultaneously improving the rendering quality. Concretely, we achieve this by deforming the vertices of a low-resolution mesh and attaching the Gaussians to a coupled high-resolution mesh.

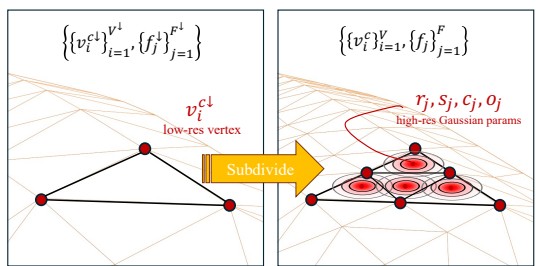

Figure 2: **Multi-resolution Gaussians-on-Mesh representation.** We use a low-res mesh for faster animation and simpler geometry and attach Gaussians on a high-res mesh for better rendering.

Formally, we define the coupled-multi-resolution Gaussians-on-Mesh representation in the canonical space as follows:

$$\text{GoM}^c \triangleq \left\{ \{v_i^{c\downarrow}\}_{i=1}^{V^\downarrow}, \{w_i^\downarrow\}_{i=1}^{V^\downarrow}, \{f_j^\downarrow\}_{j=1}^{F^\downarrow}, \{v_i^c\}_{i=1}^{V}, \{f_j\}_{j=1}^{F} \right\}. \tag{3}$$

Here, $\{v_i^{c\downarrow}\}_{i=1}^{V^\downarrow}$ and $\{f_j^\downarrow\}_{j=1}^{F^\downarrow}$ define the $V^\downarrow$ vertices and $F^\downarrow$ faces of the low-resolution mesh respectively. Note, $f_j^\downarrow \triangleq (\{\Delta_{j,k}^\downarrow\}_{k=1}^3)$, where $\Delta_{j,k}^\downarrow \in \{1, \dots, V^\downarrow\}$ is the $k$-th vertex index of the $j$-th triangle in the low-resolution mesh. To articulate it to any given human pose, we utilize linear blend skinning weights $w_i^\downarrow \in \mathbb{R}^J$ corresponding to the $i$-th vertex $v_i^\downarrow$ in the low-resolution mesh.

The high-resolution mesh is specified via $\{v_i^c\}_{i=1}^V$ and $\{f_j\}_{j=1}^F$, which subsume the $V$ vertices and $F$ faces. These are obtained by subdividing the low-resolution mesh. Different from the low-resolution mesh representation, we attach Gaussians to the high-resolution face $f_j$, i.e.,

$$f_j \triangleq (r_j, s_j, c_j, o_j, \{\Delta_{j,k}\}_{k=1}^3), \tag{4}$$

with $j \in \{1, \dots, F\}$. Here, $r_j \in so(3)$ and $s_j \in \mathbb{R}^3$ are the rotation and scale in the faces's *local* coordinate system. Moreover, $c_j \in \mathbb{R}^3$ is the RGB color, $o_j$ is the offset defined in the faces's *local* coordinate system, and $\{\Delta_{j,k}\}_{k=1}^3$ are the three vertex indices belonging to the $j$-th triangle, i.e., $\Delta_{j,k} \in \{1, \dots, V\}$. We illustrate the representation in Fig. 2.

### 3.2.2 ARTICULATION

It remains to answer 1) how we transform the defined coupled-multi-resolution Gaussians-on-Mesh representation to the target pose; and 2) how we perform rendering. To answer the first question, given a target pose $P_{\text{tg}}$, we articulate the canonical coupled-multi-resolution Gaussians-on-Mesh representation $\text{GoM}^c$ to a Gaussians-on-Mesh representation $\text{GoM}^o \triangleq \left\{ \{v_i^{o\downarrow}\}_{i=1}^{V^\downarrow}, \{f_j^\downarrow\}_{j=1}^{F^\downarrow}, \{v_i^o\}_{i=1}^{V}, \{f_j\}_{j=1}^{F} \right\}$ in the pose space utilizing linear blend skinning. Note that this representation is still multi-resolution because linear blend skinning is performed in the low-resolution space for efficiency reasons while high-quality rendering requires high-resolution Gaussian information. Concretely, we transform the canonical low-resolution 3D vertex coordinates $v_i^{c\downarrow}$ to posed low-resolution 3D vertex coordinates

$$v_i^{o\downarrow} = \texttt{LBS}\left(v_i^{c\downarrow}, w_i^\downarrow, P_{\text{tg}}\right) = \frac{\sum_{j=1}^J w_i^{j\downarrow}(R_j^p v_i^{c\downarrow} + t_j^p)}{\sum_{k=1}^J w_i^{k\downarrow}}. \tag{5}$$

Here, $\texttt{LBS}$ refers to classic linear blend skinning. Since the high-resolution canonical space mesh $\left\{\{v_i^c\}_{i=1}^V, \{f_j\}_{j=1}^F,\right\}$ is obtained from the low-resolution canonical space mesh via subdivision, it is straightforward to transfer the vertex transformations between the posed low-resolution 3D vertex coordinates $v_i^{o\downarrow}$ and its canonical counterpart $v_i^{c\downarrow}$ to the high-resolution mesh and obtain $\{v_i^o\}_{i=1}^V$.

### 3.2.3 RENDERING WITH GAUSSIAN SPLATTING

Given the pose space Gaussians-on-Mesh representation $\text{GoM}^o$ and the target camera parameters $E_{\text{tg}}$ and $K_{\text{tg}}$, we render the target image $I_{\text{tg}}^{\text{pred}}$ and the mask $M_{\text{tg}}^{\text{pred}}$ with Gaussian splatting.

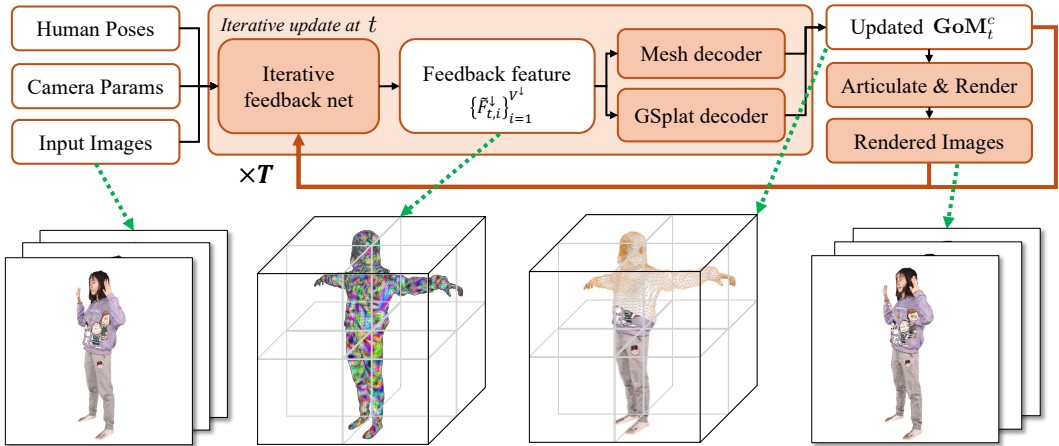

Figure 3: **Iterative feedback.** We iteratively update in a feed-forward way the vertices of the low-resolution mesh and the Gaussian parameters attached to the high-resolution mesh. We repeat the update for $T$ steps. Each step $t$ operates on the source images, camera parameters and human poses, as well as the last iteration's results including the canonical representation $\text{GoM}_{t-1}^c$ and the predicted source images rendered by $\text{GoM}_{t-1}^c$ (the brown arrows).

Our Gaussian parameters defined in Eq. (4) are located in the triangle's local coordinates. To render the images, we first transform the local Gaussian parameters to the world coordinates. Following Go-MAvatar (Wen et al., 2024), we denote the local-to-world transformation of the $j$-th high-resolution face as $A_j$. The mean of the Gaussian and its covariance are computed via

$$\mu_j = \frac{1}{3}\sum_{k=1}^{3} v_{\Delta_{j,k}}^o + A_j \cdot o_j \qquad \text{and} \qquad \Sigma_j = A_j(R_j S_j S_j^T R_j^T)A_j^T, \tag{6}$$

where $R_j$ and $S_j$ are the matrices encoding rotation $r_j$ and scale $s_j$. The color of the Gaussian is $c_j$.

### 3.3 RECONSTRUCTION WITH ITERATIVE FEEDBACK

It remains to answer how to reconstruct the canonical space coupled-multi-resolution Gaussians-on-Mesh representation $\text{GoM}^c$. For this, our `Reconstructor` defined in Eq. (2) uses sparse source images $\{I_n\}_{n=1}^N$ and masks $\{M_n\}_{n=1}^N$. Note that the sparse inputs can be multiview images or multi-frame images sampled from a monocular video, where human poses are not necessarily identical across frames. We also assume that human poses $\{P_n\}_{n=1}^N$ and camera parameters $\{E_n\}_{n=1}^N, \{K_n\}_{n=1}^N$ are given which can be human-annotated or predicted from off-the-shelf tools.

We reconstruct the canonical representation in T-pose rather than the input poses, enabling animation without any post-processing and allowing the model to handle images in unaligned poses. The added difficulty due to this choice: the gap between the canonical pose and the input poses. While scene-specific methods refine the canonical representation with gradient-based optimization, generalizable approaches must predict it in a feed-forward pass which leads to undesired reconstruction quality. To address this challenge, we propose iterative feedback updates that successively 'refine' the canonical representation in a feed-forward manner, as illustrated in Fig. 3.

To compute $\text{GoM}^c$, we perform a $T$ step iterative feedback update. We use $\text{GoM}_t^c$ to denote the output representation from the $t$-th step, i.e., $t \in \{0, \dots, T\}$ and let

$$\text{GoM}_t^c \triangleq \left\{ \{v_{t,i}^{c\downarrow}\}_{i=1}^{V^\downarrow}, \{w_i^\downarrow\}_{i=1}^{V^\downarrow}, \{f_j^\downarrow\}_{j=1}^{F^\downarrow}, \{v_{t,i}^c\}_{i=1}^V, \{f_{t,j}\}_{j=1}^F \right\}. \tag{7}$$

Here, the step-dependent face information is given by

$$f_{t,j} \triangleq (r_{t,j}, s_{t,j}, c_{t,j}, o_{t,j}, \{\Delta_{j,k}\}_{k=1}^3), \quad \text{with} \quad j \in \{1, \dots, F\}. \tag{8}$$

Note, $\text{GoM}_0^c$, the canonical representation at $t = 0$, is the initialization and $\text{GoM}^c = \text{GoM}_T^c$.

We emphasize that our iterative feedback updates the low-resolution mesh vertices $\{v_i^{c\downarrow}\}_{i=1}^{V^\downarrow}$, and the Gaussian parameters $\{r_j, s_j, c_j, o_j\}_{j=1}^F$ associated with the high-resolution faces. The vertices in

the high-resolution mesh $\{v_{t,i}^c\}_{i=1}^V$ follow the low-resolution update, analogously to the articulation update discussed in Section 3.2.2.

At each step $t$, we update the low-resolution mesh vertices and high-resolution Gaussian parameters using the following equations:

$$v_{t,i}^{c\downarrow} = v_{t-1,i}^{c\downarrow} + \text{MLP}(\tilde{F}_{t,i}^{\downarrow}), \tag{9}$$

$$r_{t,j}, s_{t,j}, c_{t,j}, o_{t,j} = \text{MLP}(\text{cat}(\tilde{F}_{t,j}, \{F_{n,t,j}\}_{n=1}^N)). \tag{10}$$

Here, $\tilde{F}_{t,i}^{\downarrow}, i \in \{1, \ldots, V^{\downarrow}\}$ is our 'feedback' feature for the $i$-th vertex in the low-resolution mesh. Further, $\tilde{F}_{t,j}, j \in \{1, \ldots, F\}$ in Eq. (10) is a 'feedback' feature for the $j$-th face in the high-resolution mesh. It is acquired by first interpolating $\tilde{F}_{t,i}^{\downarrow}, i \in \{1, \ldots, V^{\downarrow}\}$ to get vertex features in the high-resolution mesh and then concatenating the 3 vertices' features belonging to the $j$-th face. To preserve details, we also concatenate source image features $\{F_{n,t,j}\}_{n=1}^N$ which are obtained by projecting the mean of the $j$-th Gaussian at step $t$ to the $n$-th view.

To compute the 'feedback' feature $\tilde{F}_{t,i}^{\downarrow}, i \in \{1, \ldots, V^{\downarrow}\}$, we first render the source views using the canonical representation from the last iteration via

$$I_{n,t-1}^{\text{pred}}, M_{n,t-1}^{\text{pred}} = \texttt{Renderer}(\text{GoM}_{t-1}^c, P_n, E_n, K_n), \quad n \in \{1, \ldots, N\}. \tag{11}$$

Then we extract image features from $\{I_{n,t-1}^{\text{pred}}\}_{n=1}^N$. For each vertex $v_{t,i-1}^{c\downarrow}$ in the low-resolution mesh, we extract pixel-aligned source image features and predicted image features. We concatenate both and feed them into an iterative feedback network. The iterative feedback network consists of a multi-source fusion block that mixes the information from $N$ sources, and a Point Transformer that encodes all the vertices. Its output feature is $\{\tilde{F}_{t,i}^{\downarrow}\}_{i=1}^{V^{\downarrow}}$. Please refer to Appendix B for more details.

## 3.4 Training

Both rendering and reconstruction using our iterative feedback network and coupled representation are end-to-end differentiable. To learn the network parameters, we use a training loss composed of L1 and perceptual losses, comparing predicted and ground-truth RGB images, L1 loss for masks as well as a Laplacian loss for regularization. The loss is averaged over all source and target images, as well as all $T$ iterative feedback steps. Formally, we minimize the average of $\text{Loss}_{n,t}, n \in \{1, \ldots, N, \text{tg}\}, t \in \{1, \ldots, T\}$ and

$$\text{Loss}_{n,t} = L_1(I_n, I_{n,t}^{\text{pred}}) + \lambda_{\text{per}}\text{Perceptual}(I_n, I_{n,t}^{\text{pred}}) + \lambda_M L_1(M_n, M_{n,t}^{\text{pred}}) + \lambda_{\text{lap}}\text{Lap}(\text{GoM}_t^c). \tag{12}$$

Here, $L_1(\cdot, \cdot)$ is the L1 loss. Perceptual$(\cdot, \cdot)$ is the perceptual loss between predictions and ground-truths, e.g., SSIM or LPIPS. Lap$(\cdot)$ is the Laplacian loss applied on the low-resolution mesh of the canonical GoM representation. $\lambda_{\text{per}}$, $\lambda_M$ and $\lambda_{\text{lap}}$ are user-specified hyperparameters.

## 4 Experiments

### 4.1 Implementation details

**Representation details.** We initilize the low-resolution mesh $\left\{\{v_{0,i}^{c\downarrow}\}_{i=1}^{V^{\downarrow}}, \{f_j^{\downarrow}\}_{j=1}^{F^{\downarrow}}\right\}$ in $\text{GoM}_0^c$ with SMPL or SMPL-X, depending on the human pose representation used in the dataset. The high-resolution mesh is obtained by subdividing the low-resolution mesh.

**Architecture details.** We provide the detailed architecture in Appendix B.

**Training details.** We set $\lambda_{\text{per}} = 1.0$, $\lambda_M = 5.0$ and $\lambda_{\text{lap}} = 100$ in Eq. (12) on THuman2.0 and $\lambda_{\text{per}} = 1.0$, $\lambda_M = 0$ and $\lambda_{\text{lap}} = 100$ in Eq. (12) on AIST++. We use the SSIM loss in THuman2.0 and the LPIPS loss in AIST++ following the baselines. We use Adam as the optimizer. On THuman2.0, the learning rates of the image encoder and the rest of the model are $1e-4$ and $5e-5$ respectively. On AIST++, we set the learning rate of all parameters to $5e-5$. We optimize the model for 200K iterations on THuman2.0 and 100K iterations on AIST++.

Table 1: **Comparison on THuman2.0.** The proposed method improves state-of-the-art in PSNR, LPIPS* and FID. We highlight the best result in bold font. Methods marked in gray are per-scene optimized methods.

| Number of source views | Method | PSNR↑ | LPIPS*↓ | FID↓ |
|---|---|---|---|---|
| 3 | GoMAvatar (Wen et al., 2024) | 23.05 | 133.98 | 87.51 |
| | 3DGS-Avatar (Qian et al., 2024b) | 21.25 | 160.48 | 157.21 |
| | iHuman (Paudel et al., 2024) | 22.77 | 131.67 | 101.70 |
| | NHP (Kwon et al., 2021) | 23.32 | 184.69 | 136.56 |
| | NIA (Kwon et al., 2023) | 23.20 | 181.82 | 127.30 |
| | GHG (Kwon et al., 2024) | 21.90 | 133.41 | 61.67 |
| | LIFe-GoM (Ours) | **24.65** | **110.82** | **51.27** |
| 5 | GPS-Gaussian (Zheng et al., 2024) | 20.39 | 152.34 | 65.90 |
| | LIFe-GoM (Ours) | **25.57** | **105.39** | **38.57** |

Table 2: **Comparison on XHuman.** We evaluate on XHuman to prove the ability of cross-domain generalization. The proposed method improves state-of-the-art in PSNR, LPIPS* and FID. We highlight the best result in bold font.

| Method | PSNR↑ | LPIPS*↓ | FID↓ |
|---|---|---|---|
| GHG (Kwon et al., 2024) | 23.52 | 112.91 | 50.51 |
| LIFe-GoM (Ours) | **25.32** | **99.32** | **42.90** |

## 4.2 EXPERIMENTAL SETUP

We evaluate our approach in two settings: 1) **Multiview source images.** Our approach can take multiview images as input to produce a canonical representation; 2) **Multi-frame source images.** Since our approach directly learns a 3D representation in the canonical space instead of a posed space, our method can also operate on images showing various human poses, e.g., frames sampled from a monocular video. Our approach can synthesize both novel views and novel poses.

**Datasets.** We validate our approach on THuman2.0 (Yu et al., 2021), XHuman (Shen et al., 2023) and AIST++ (Li et al., 2021) quantitatively. We use THuman2.0 to evaluate our approach in the setting of multiview source images. XHuman is used to validate the cross-domain generalization of our approach. In other words, we train our model on THuman2.0 and test on XHuman without fine-tuning. The AIST++ dataset is used to evaluate the multi-frame source image setting. Please see Appendix C for detailed dataset setup.

**Baselines.** We compare with GoMAvatar (Wen et al., 2024), 3DGS-Avatar (Qian et al., 2024b), iHuman (Paudel et al., 2024), NHP (Kwon et al., 2021), NIA (Kwon et al., 2023), GHG (Kwon et al., 2024) and GPS-Gaussian (Zheng et al., 2024) on THuman2.0. On AIST++, we compare with HumanNeRF (Weng et al., 2022), GoMAvatar (Wen et al., 2024), 3DGS-Avatar (Qian et al., 2024b), iHuman (Paudel et al., 2024) and ActorsNeRF (Mu et al., 2023). Please refer to Appendix D for details.

**Evaluation metrics.** We report PSNR, LPIPS*(= LPIPS × 1000) and FID on THuman2.0 following GHG (Kwon et al., 2024). We report PSNR, SSIM and LPIPS* on AIST++ following ActorsNeRF (Mu et al., 2023).

## 4.3 QUANTITATIVE RESULTS

**THuman2.0.** We summarize our results in Table 1 for both the three-view and the five-view setting.

In the three-view setting, our method significantly outperforms per-scene optimized methods including GoMAvatar, 3DGS-Avatar and iHuman, and generalizable approaches including NHP, NIA, and GHG in PSNR, LPIPS*, and FID. Our approach achieves 24.65/110.82/51.27 in PSNR/LPIPS*/FID, compared to GHG's 21.90/133.41/61.67. Importantly, we use 330K Gaussians for splatting, $7.5\times$ fewer than GHG's 2.8M, resulting in faster rendering (10.52ms vs. GHG's 20.30ms) at $1024 \times 1024$ resolution on a NVIDIA A100 GPU. Our method takes 907.92ms to reconstruct the coupled-multi-resolution Gaussians-on-Mesh in canonical space, significantly faster than scene-specific methods but slower than GHG. That said, reconstruction only needs to be done once per input subject, as the reconstructed avatar will be cached and reused for articulation and rendering, which runs at 95 FPS.

Table 3: **Comparison on AIST++.** We achieve comparable quality as ActorsNeRF while requiring much less time in reconstruction or optimization. We highlight the best result in bold font. Methods marked in gray are per-scene optimized methods.

| Method | PSNR↑ | SSIM↑ | LPIPS*↓ | Reconstruction or optimization time↓ |
|---|---|---|---|---|
| HumanNeRF (Weng et al., 2022) | 24.21 | 0.9760 | 29.66 | ∼2h |
| GoMAvatar (Wen et al., 2024) | 24.34 | 0.9780 | 25.34 | ∼10h |
| 3DGS-Avatar (Qian et al., 2024b) | 25.14 | 0.9784 | 27.17 | ∼2min |
| iHuman (Paudel et al., 2024) | 25.17 | 0.9805 | 22.90 | 6.61s |
| ActorsNeRF (Mu et al., 2023) | 25.23 | 0.9809 | 22.45 | ∼4h |
| LIFe-GoM (Ours) | **25.25** | **0.9812** | **21.61** | **589.27ms** |

(a) Reference image  (b) Novel view synthesis

Figure 4: **Cross-domain generalization** on DNA-Rendering dataset w/o finetuning.

(a) Reference image  (b) Novel pose synthesis

Figure 5: **Novel pose synthesis.** Poses are from BEDLAM dataset.

We compare our approach to GPS-Gaussian using five images. As GPS-Gaussian relies on depth prediction between adjacent views, five images are the minimum it needs. Despite that, it still fails in non-overlapping regions. Our approach significantly improves upon GPS-Gaussian in this setting.

**XHuman.** We summarize the cross-dataset generalization results in Table 2. We directly apply GHG and our approach trained on THuman2.0 in the setting of 3 source views to the XHuman dataset without any finetuning. Our approach achieves PSNR/LPIPS*/FID of 25.32/99.32/42.90, significantly outperforming GHG's 23.52/112.91/50.51.

**AIST++.** Table 3 summarizes quantitative results on AIST++. Our method achieves 25.25/0.9812/21.61 in PSNR/SSIM/LPIPS*, matching ActorsNeRF's 25.23/0.9809/22.11 and surpassing per-scene optimized methods. Importantly, our method needs only 589 ms for 3D reconstruction, whereas iHuman, the fastest scene-specific method, requires 6.61s and other baselines take minutes to hours.

### 4.4 QUALITATIVE RESULTS

Please refer to Appendix E for more qualitative results, including a comparison to baselines.

**Cross-domain generalization.** We demo our approach on cross-domain generalization in Fig. 4, using the DNA-Rendering data (Cheng et al., 2023). Without fine-tuning, our approach can generalize to challenging subjects, e.g., loose clothes.

**Novel pose synthesis.** Instead of directly reconstructing human avatars in the pose of the source images, our approach outputs the canonical representation in T-pose via the `Reconstructor`. Benefitting from this choice, we can synthesize novel poses without postprocessing such as binding the skeletons. In Fig. 5, we retarget the avatar to challenging new pose sequences from the BEDLAM dataset (Black et al., 2023). The avatar is reconstructed using the model which was used to report results in the 3 source view setting of Table 1.

### 4.5 ABLATION STUDIES

**Analysis of iterative step choice.** We study how the number of iterations ($T$) influences the reconstruction time and rendering quality. Results are summarized in Table 4 and Fig. 6(a). Note that $T = 1$ means a single feed-forward pass, i.e., iterative updates are disabled. Using more iterations improves the rendering quality at the expense of more reconstruction time (∼290ms per iteration). The PSNR improves by $+0.78$ and $+0.91$ when $T = 2$ and $T = 3$ respectively compared to $T = 1$. Starting with $T = 4$, the benefit of more iterations diminishes. We choose $T = 3$ in our final model to balance rendering quality and reconstruction time.

Table 4: **Iterative step choice.** More iterations lead to better rendering at the expense of longer reconstruction. We use 3 iterations for the best quality-speed tradeoff, as highlighted in gray.

| # iterations | PSNR↑ | LPIPS*↓ | FID↓ | Reconstruction time (ms)↓ |
|:---:|:---:|:---:|:---:|:---:|
| 1 | 23.74 | 124.58 | 64.59 | 328.79 |
| 2 | 24.52 | 112.47 | 52.16 | 618.67 |
| 3 | 24.65 | 110.82 | 51.27 | 907.92 |
| 4 | 24.69 | 110.46 | 51.25 | 1198.14 |
| 5 | 24.70 | 110.38 | 51.02 | 1563.92 |

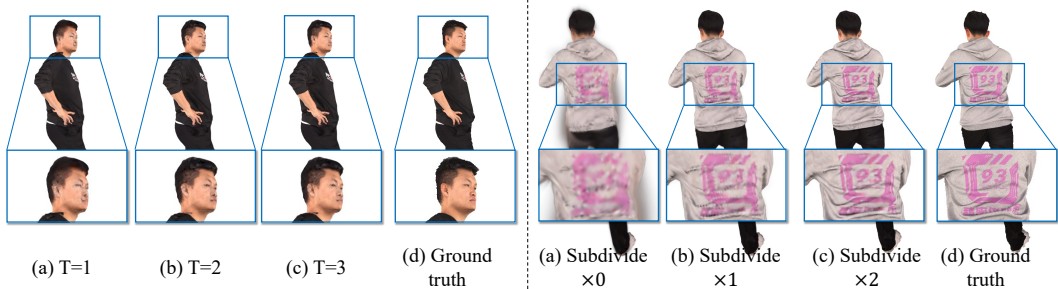

| (a) T=1 | (b) T=2 | (c) T=3 | (d) Ground truth | (a) Subdivide ×0 | (b) Subdivide ×1 | (c) Subdivide ×2 | (d) Ground truth |

Figure 6: **Ablation studies.** We study the effect of iterative feedback (left). The geometry improves as the number of iterations increases. We show the importance of linking Gaussians to the high-resolution mesh (right). The high-resolution mesh is subdivided from the low-resolution counterpart. A higher resolution yields better texture details.

Table 5: **Coupled-multi-resolution Gaussians-on-Mesh.** Increasing the number of subdivisions improves rendering quality at the cost of longer reconstruction and rendering times. We subdivide twice in our final model to ensure quality while maintaining real-time, as highlighted in gray.

| # subdivision | PSNR↑ | LPIPS*↓ | FID↓ | Reconstruction time (ms)↓ | Rendering time (ms)↓ |
|:---:|:---:|:---:|:---:|:---:|:---:|
| 0 | 24.76 | 140.60 | 93.44 | 538.02 | 3.20 |
| 1 | 24.88 | 118.64 | 58.45 | 607.49 | 3.93 |
| 2 | 24.65 | 110.82 | 51.27 | 907.92 | 10.52 |

**Coupled-multi-resolution Gaussians-on-Mesh.** As mentioned in Section 3.2.1 and Section 3.3, we update the vertices of the low-resolution mesh, while the Gaussians are associated with the high-resolution mesh. Both are updated jointly. This choice is necessary for two reasons: 1) simply updating the vertices of the high-resolution mesh increases the reconstruction time from 907.92ms to 12.45s, making it too slow for both training and inference; 2) learning Gaussians in the high-resolution mesh guarantees good rendering quality. Note that the high-resolution mesh is obtained by subdividing the low-resolution mesh. In Table 5, we show that the rendering improves to 118.64/58.45 and 110.82/51.27 in LPIPS*/FID when subdividing once and twice respectively from 140.60/93.44 without subdivision. The improvement can also be observed in Fig. 6(b). Note that we do not observe consistent improvement in PSNR. This is because PSNR sometimes prefers blurry results. The resolution of the high-resolution mesh affects both the reconstruction speed and the rendering speed since we render the source images during the reconstruction stage. As the reconstruction time is still less than 1s, we choose to subdivide twice for better rendering quality.

## 5 CONCLUSIONS

We tackle the problem of generalizable reconstruction of an animatable human avatar from sparse inputs. We propose a feed-forward network featuring iterative updates with iterative feedback and coupled-multi-resolution Gaussians-on-Mesh representation. Our method achieves state-of-the-art rendering quality. It requires less than 1s for avatar reconstruction and renders at 95 FPS.

**Acknowledgements.** Work supported in part by NSF grants 2008387, 2045586, 2106825, MRI 1725729, and NIFA award 2020-67021-32799.

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

APPENDIX — LIFE-GOM: GENERALIZABLE HUMAN RENDERING WITH LEARNED ITERATIVE FEEDBACK OVER MULTI-RESOLUTION GAUSSIANS-ON-MESH

This appendix is structured as follows:

- Sec. A summarizes mesh representations in human modeling;
- Sec. B provides the detailed architecture of the iterative feedback module;
- Sec. C details the datasets;
- Sec. D shows baseline details on the presented datasets;
- Sec. E provides additional results and analysis. Please visit the project webpage[1] for more qualitative results;
- Sec. F showcases failure cases in our approach.

## A  ADDTIONAL RELATED WORKS

**Mesh representations in human modeling.** Meshes as an explicit representation are easy to animate and can be rendered at a fast speed. Further, meshes can be easily integrated into the classic graphics pipeline. Therefore, meshes are widely used in human modeling (Liao et al., 2024; Zhang et al., 2023; Liao et al., 2023). However, as mentioned in GoMAvatar (Wen et al., 2024), it is difficult to learn to deform the mesh using photometric losses and mesh rasterization. Hence, methods using meshes as the underlying representation either extract them from other types of representations such as a signed distance function (SDF) (Zhang et al., 2023; Liao et al., 2023), or apply explicit supervision on the geometry, e.g., supervising surface normals (Liao et al., 2024; Zhang et al., 2023; Liao et al., 2023). In contrast, we opt to use the Gaussians-on-Mesh representation that binds Gaussians on the mesh and uses Gaussian splatting for rendering. This enables us to overcome the difficulty in optimization. Consequently, our entire model is learned via photometric losses only. Further, Gaussians-on-Mesh leverages the flexibility of Gaussian Splatting, enabling more photorealistic rendering than textured meshes.

## B  DETAILS FOR THE ITERATIVE FEEDBACK MODULE

The detailed architecture of the iterative feedback module is provided in Fig. 7. Given source images and rendered images, we first extract image features via an image encoder. Then we apply multi-source fusion which samples aligned image features for each of the low-resolution vertices $\{v_{t-1,i}^{c\downarrow}\}_{i=1}^{V^\downarrow}$ and mixes the features from $N$ sources. After that, a Point Transformer is adopted to encode all vertices. Note that the iterative feedback module operates on the low-resolution mesh.

**Image encoder.** We use ResNet-18 (He et al., 2016) with ImageNet pretrained weights as the image encoder. The image feature is the concatenation of features from 5 intermediate layers and therefore has a dimension of 1192, i.e., $C = 1192$ in Fig. 7. Concatenating multi-level features ensures a large receptive field and is essential for iterative updates.

**Multi-source fusion.** Multi-source fusion first samples image features for all vertices in the low-resolution mesh. Concretely, the $i$-th vertex $v_i^{c\downarrow}, i \in \{1, \ldots, V^\downarrow\}$ is first articulated via the available source human poses $\{P_n\}_{n=1}^N$ and then projected onto images via the available camera intrinsics $\{K_n\}_{n=1}^N$ and extrinsics $\{E_n\}_{n=1}^N$. The aligned features are sampled at the projected points from each of the $N$ source images. Subsequently we mix the sampled features from the $N$ source images using two Transformer encoder layers. The query matrix $Q$, key matrix $K$ and value matrix $V$ for each Transformer encoder are illustrated in Fig. 8. The input, intermediate and output dimensions are $C = 1192$. We use 6 heads in the attention layers. Note that in the second Transformer encoder layer, we use a learnable vertex embedding $\{e_{t-1,i}^\downarrow\}_{i=1}^{V^\downarrow}$ as the query. The learnable vertex embedding is updated in iterative updates together with the low-resolution vertices.

---

[1] https://wenj.github.io/LIFe-GoM/

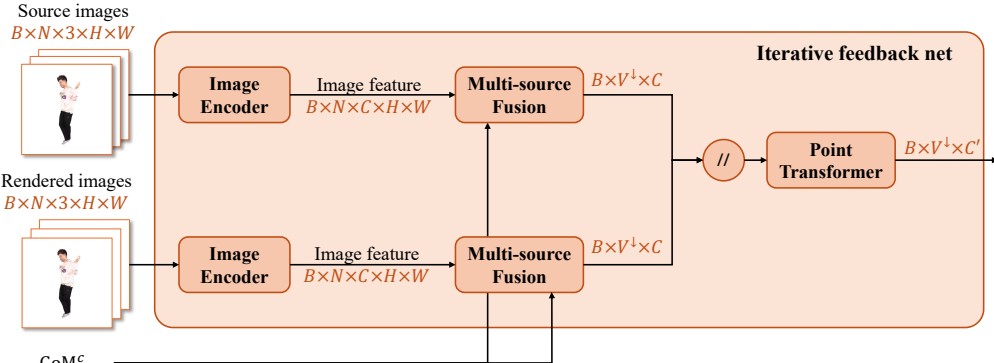

Figure 7: **Iterative feedback module.** The iterative feedback module takes as input the representation $GoM_{t-1}^c$ obtained from the previous iteration, the source images and images rendered with $GoM_{t-1}^c$. The module is designed to compare the rendered images and source images, and to summarize the result in a feature vector of dimension $C'$ for each vertex in the low-resolution mesh. Here, $B$ denotes the batch size, $N$ refers to the number of source images, and $H$ and $W$ are the height and weight of the images respectively. Further, $V^\downarrow$ is the number of vertices in the *low*-resolution mesh, $C$ refers to the dimension of the feature vector from the image encoder, and $C'$ denotes the dimension of the output feature from the Point Transformer. The entire module operates on the low-resolution mesh.

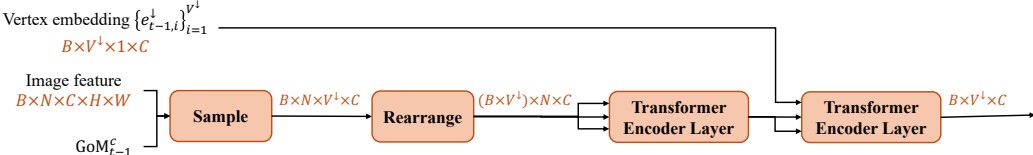

Figure 8: **Multi-source fusion.** Multi-source fusion first samples the vertex-aligned image features from the encoded images. Then we use two Transformer encoder layers to fuse the information from each of the $N$ source images. In the Transformer encoder layers, the three input arrows from top to bottom represent the query matrix $Q$, the key matrix $K$, and the value matrix $V$ of the attention layer respectively. We additionally associate a learnable vertex embedding with each vertex. Please check Appendix B for details.

**Point Transformer.** The Point Transformer (Zhao et al., 2021) is used to encode the vertices and to produce high-level features for all low-resolution vertices. The output dimension of each vertex is 32, i.e., $C' = 32$ in Fig. 7.

## C    DATASET DETAILS

**THuman2.0 (Yu et al., 2021)**. We use THuman2.0 to evaluate our approach in the setting of multiview source images. THuman2.0 has 526 high-quality 3D human scans, texture maps and corresponding SMPL-X parameters. We follow the experimental setup of GHG (Kwon et al., 2024) and split the dataset into 426 subjects for training and 100 subjects for evaluation. We render multiview images from the 3D scans. 3 or 5 images are used as source images and the remaining ones are used for evaluation.

**XHuman (Shen et al., 2023)**. We use XHuman to validate our approach to cross-domain generalization quantitatively. The dataset provides 20 subjects with high-quality scans and SMPL-X parameters. We sample three scans (f00001, f00051, f00101) for each subject. We prepare the dataset in the same way as THuman2.0. To validate the ability of cross-domain generalization, we only evaluate in this dataset without any finetuning.

**AIST++ (Li et al., 2021).** The AIST++ dataset is used to evaluate the setting of multi-frame source images. The AIST++ dataset consists of multiview dancing videos, camera calibration parameters, and human motions represented in SMPL poses. We adopt the training and evaluation protocol of ActorsNeRF (Mu et al., 2023). Specifically, we use subjects 1-15 and 21-30 for training and leave out subjects 16-20 for evaluation. We choose one motion sequence for each subject. We only use camera 1 for training. During evaluation, we sample 5 source images from Camera 1 and use Camera 2-7 to evaluate generalizable novel view and pose synthesis.

## D    BASELINE DETAILS

We compare with per-scene optimized approaches including GoMAvatar (Wen et al., 2024), 3DGS-Avatar (Qian et al., 2024b) and iHuman (Paudel et al., 2024), and other generalizable human rendering approaches including NHP (Kwon et al., 2021), NIA (Kwon et al., 2023), GHG (Kwon et al., 2024) and GPS-Gaussian (Zheng et al., 2024) on THuman2.0. We use 3 source images when comparing with GoMAvatar, 3DGS-Avatar, iHuman, NHP, NIA and GHG. For the comparison with GPS-Gaussian, we adopt 5 source images following the setting of GHG (Kwon et al., 2024), since GPS-Gaussian requires the source views to overlap with each other and thus does not work well with very sparse views. We compared with the pretrained GHG (Kwon et al., 2024) on XHuman. On AIST++, we compare with HumanNeRF (Weng et al., 2022), GoMAvatar (Wen et al., 2024), 3DGS-Avatar (Qian et al., 2024b), iHuman (Paudel et al., 2024) and ActorsNeRF (Mu et al., 2023). HumanNeRF, GoMAvatar, 3DGS-Avatar and iHuman need to be trained per scene. ActorsNeRF adopts a two-stage training: In the first stage, it learns a categorical prior from large-scale datasets. In the second stage, it adopts per-scene optimization given the source images. Now we detail the training setup of each baseline.

**NHP (Kwon et al., 2021), NIA (Kwon et al., 2023) and GHG (Kwon et al., 2024).** We follow the same setting as reported in GHG (Kwon et al., 2024) for training and evaluation.

**GPS-Gaussian (Zheng et al., 2024).** As described in GHG (Kwon et al., 2024), GPS-Gaussian can work on as few as 5 input views. We render the THuman2.0 dataset to accommodate this setting. We use the default parameters provided in GPS-Gaussian to train the model.

**GoMAvatar (Wen et al., 2024).** GoMAvatar originally takes SMPL parameters as inputs. On THuman2.0, we adjust it to work with SMPL-X parameters. On AIST++, we keep the original setting and use SMPL. We train the model for 100K iterations on both datasets instead of 200K iterations as stated in the paper, to avoid overfitting sparse inputs.

**3DGS-Avatar (Qian et al., 2024b).** Similar to GoMAvatar, we adapt 3DGS-Avatar to take SMPL-X parameters as inputs on THuman2.0 and keep its original setting on AIST++. We train it for 2K iterations to avoid overfitting the very sparse inputs.

**iHuman (Paudel et al., 2024).** We adapt iHuman to work with SMPL-X parameters on the THuman2.0 dataset. It requires subdivided SMPL or SMPL-X templates as inputs. For a fair comparison, we adopt the same subdivision strategy for the SMPL-X template as ours, in which we subdivide all faces twice. We find that the model cannot converge after 15 epochs, the number of epochs specified in the original iHuman paper. Instead, we use 150 epochs which takes a longer time for training but provides better rendering quality. On AIST++, we use the default hyperparameters for training.

**HumanNeRF (Weng et al., 2022) and ActorsNeRF (Mu et al., 2023).** We follow the same setting as reported in ActorsNeRF (Mu et al., 2023) for training and evaluation.

## E    ADDITIONAL ANALYSIS

For additional qualitative results we refer the reader to the project webpage. It contains videos for freeview rendering, cross-domain generalization and novel pose synthesis.

### E.1    ADDITIONAL COMPARISONS ON THUMAN2.0

We compare our approach against GoMAvatar (Wen et al., 2024), iHuman (Paudel et al., 2024), and GHG (Kwon et al., 2024) in the setting of 3 source images on THuman2.0. Note that GoMAvatar

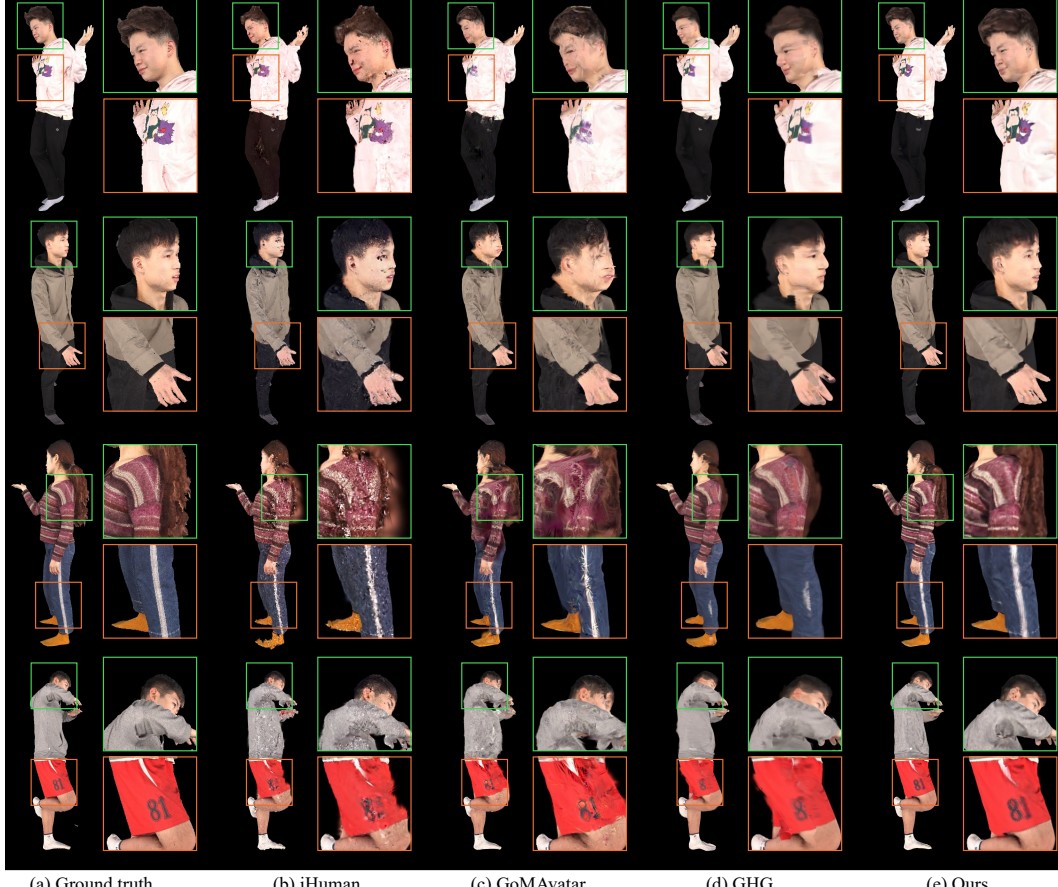

(a) Ground truth      (b) iHuman      (c) GoMAvatar      (d) GHG      (e) Ours

Figure 9: **Comparison to baselines in the setting of 3 source images on THuman2.0.** Our method produces less noise than iHuman and GoMAvatar, and more accurate geometry and sharper details than GHG.

and iHuman are scene-specific methods while GHG is a generalizable approach, the same as ours. In Fig. 9, we showcase additional qualitative comparisons to the baselines.

In the setting of very sparse inputs, e.g., 3 views, scene-specific methods suffer from overfitting and struggle to render uncorrupted novel views. Generalizable approaches, in contrast, constrain the output space with the data priors learned from large-scale datasets, which leads to more plausible rendering quality. Compared to other generalizable approaches, ours outputs more accurate geometry and sharper details.

We show the qualitative comparison between ours and GPS-Gaussian in the setting of 5 views on THuman in Fig. 10. GPS-Gaussian (Zheng et al., 2024) relies on stereo depth estimation to locate the Gaussians. During inference time, it takes as inputs two adjacent views and interpolates the novel views in between. Therefore, it requires the adjacent views to overlap with each other. As mentioned in GHG (Kwon et al., 2024), 5 views are the minimal number of input views that GPS-Gaussian can work on. Even with 5 views as inputs, we still find that it fails in the non-overlapped regions, leaving incomplete silhouettes in rendering. In contrast, ours outperforms GPS-Gaussian qualitatively and quantitatively in the setting of 5 views and can work on as few as 3 views. Another key difference between GPS-Gaussian and our approach is that we reconstruct the human subject in the canonical T-pose while the representation of GPS-Gaussian is in the same pose as the source images. Therefore, ours can take images in unaligned poses as inputs and render novel poses without extra effort, as demonstrated in Fig. 5.

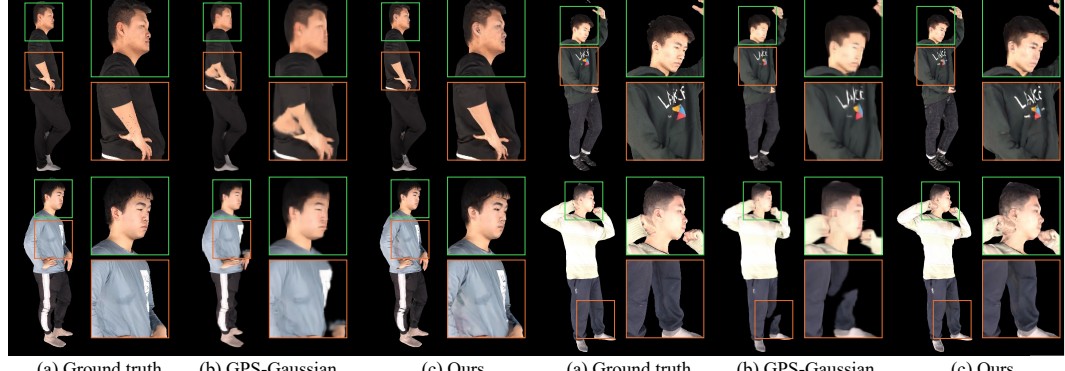

(a) Ground truth  (b) GPS-Gaussian  (c) Ours  (a) Ground truth  (b) GPS-Gaussian  (c) Ours

Figure 10: **Comparison to GPS-Gaussian in the setting of 5 source images on THuman2.0.** Our method produces more complete shape and sharper details.

Table 6: **Per-scene breakdown on AIST++.** We use lighter gray for scene-specific methods, while the others are generalizable methods.

|  | PSNR ↑ | SSIM ↑ | LPIPS* ↓ | PSNR ↑ | SSIM ↑ | LPIPS* ↓ | PSNR ↑ | SSIM ↑ | LPIPS* ↓ |
|---|---|---|---|---|---|---|---|---|---|
|  |  | d16 |  |  | d17 |  |  | d18 |  |
| HumanNeRF | 24.37 | 0.9752 | 29.59 | 24.86 | 0.9762 | 29.39 | 22.77 | 0.9738 | 33.02 |
| GoMAvatar | 24.35 | 0.9769 | 24.80 | 25.12 | 0.9780 | 25.17 | 23.18 | 0.9771 | 27.57 |
| 3DGS-Avatar | 25.22 | 0.9776 | 26.01 | 25.71 | 0.9787 | 27.70 | 23.75 | 0.9757 | 29.98 |
| iHuman | 25.41 | 0.9804 | 21.79 | 25.59 | 0.9805 | 23.69 | 24.25 | 0.9786 | 24.37 |
| ActorsNeRF | 25.22 | 0.9796 | 22.03 | 25.88 | 0.9808 | 22.85 | 24.50 | 0.9811 | 22.38 |
| Ours | 25.43 | 0.9801 | 21.48 | 25.73 | 0.9812 | 21.94 | 24.46 | 0.9810 | 22.21 |
|  |  | d19 |  |  | d20 |  |  | Average |  |
| HumanNeRF | 24.51 | 0.9759 | 28.68 | 24.55 | 0.9791 | 27.63 | 24.21 | 0.9760 | 29.66 |
| GoMAvatar | 24.36 | 0.9773 | 25.24 | 24.68 | 0.9806 | 23.95 | 24.34 | 0.9780 | 25.34 |
| 3DGS-Avatar | 25.32 | 0.9783 | 27.70 | 25.70 | 0.9819 | 24.44 | 25.14 | 0.9784 | 27.17 |
| iHuman | 25.11 | 0.9800 | 23.29 | 25.48 | 0.9829 | 21.37 | 25.17 | 0.9805 | 22.90 |
| ActorsNeRF | 25.24 | 0.9801 | 22.87 | 25.30 | 0.9827 | 21.34 | 25.23 | 0.9809 | 22.29 |
| Ours | 25.19 | 0.9805 | 21.42 | 25.43 | 0.9829 | 20.98 | 25.25 | 0.9812 | 21.61 |

## E.2 Additional comparisons on AIST++

Following ActorsNeRF, we also list the per-scene breakdown on 5 evaluation scenes on AIST++ in Table 6.

Next, we present the qualitative comparison to GoMAvatar, iHuman and ActorsNeRF in Fig. 11. AIST++ is challenging for two reasons: 1) The subjects perform challenging and diverse poses in the videos. 2) The poses provided by the dataset are less accurate and the masks are predicted from off-the-shelf tools which are also less accurate. Due to the explicit nature of the Gaussians-on-Mesh representation, our method produces fewer floaters than NeRF-based ActorNeRF. Meanwhile, we capture better silhouettes and produce less noise compared to iHuman and GoMAvatar, the two scene-specific methods.

## E.3 Input pose sensitivity

We quantitatively compare the sensitivity to input pose accuracy for our approach and GHG (Kwon et al., 2024). In this experiment, we add Gaussian noise of increasing standard deviation $(0.1, 0.3, 0.5)$ to the poses provided by THuman2.0. The results are summarized in Table 7. Both methods are affected by the accuracy of the input poses. However, our approach improves upon GHG in all noise levels.

To make our approach less sensitive to the accuracy of input poses, we can explore a pose refinement network that is jointly trained with the iterative feedback. We leave it for future work.

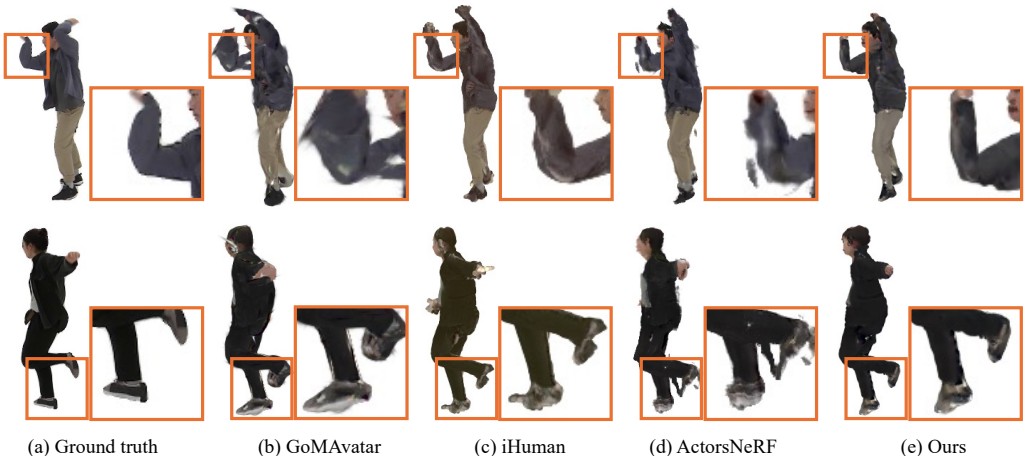

|  (a) Ground truth | (b) GoMAvatar | (c) iHuman | (d) ActorsNeRF | (e) Ours |

Figure 11: **Comparisons on baselines on AIST++.** Our method has fewer floaters compared to ActorsNeRF and produces more complete shape than GoMAvatar and iHuman. Meanwhile, ours is 11× faster than iHuman in reconstruction.

Table 7: **Comparison regarding inaccurate input poses.** We add random Gaussian noise of different standard deviations to the poses provided by THuman2.0. Our method outperforms GHG for all noise levels.

| Noise | std=0.1 | | | std=0.3 | | | std=0.5 | | |
|---|---|---|---|---|---|---|---|---|---|
| Method | PSNR↑ | LPIPS*↓ | FID↓ | PSNR↑ | LPIPS*↓ | FID↓ | PSNR↑ | LPIPS*↓ | FID↓ |
| GHG | 21.25 | 136.87 | 62.03 | 19.66 | 149.73 | 64.15 | 18.53 | 163.48 | 68.57 |
| Ours | 23.96 | 113.80 | 53.15 | 22.02 | 123.15 | 57.22 | 20.43 | 134.86 | 62.84 |

Table 8: **Importance of SMPL-X shape input.** We compare our method w/ and w/o SMPL-X shape input for mesh initialization. We only observe a slight drop in performance if the SMPL-X shape is not used.

| Method | PSNR↑ | LPIPS*↓ | FID↓ |
|---|---|---|---|
| Ours w/o SMPL-X shape | 24.15 | 112.88 | 52.01 |
| Ours w/ SMPL-X shape | 24.65 | 110.82 | 51.27 |

### E.4 INPUT SHAPE SENSITIVITY

We quantitatively evaluate the sensitivity to input SMPL-X shape accuracy for our approach in Table 8. In this experiment, we initialize the canonical mesh with the average SMPL shape by setting the beta parameter to a tensor of all zeros for all subjects. Neither ground-truth nor predicted SMPL shapes are used for any subject. We call this setting "Ours w/o SMPL-X shape" in the table.

Our method attains PSNR/LPIPS*/FID of 24.15/112.88/52.01 w/o SMPL shapes as input. Compared to 24.65/110.82/51.27 with SMPL shapes as input, we only observe a small drop, which shows the robustness of our method to the accuracy of SMPL-X shapes. Even *without* SMPL-X shapes, our method still significantly outperforms GHG's 21.90/133.41/61.67 *with* SMPL-X shapes as input. We further demonstrate the robustness in Fig. 12. Although the average shape is smaller than the ground-truth shape, our method still captures the correct shape.

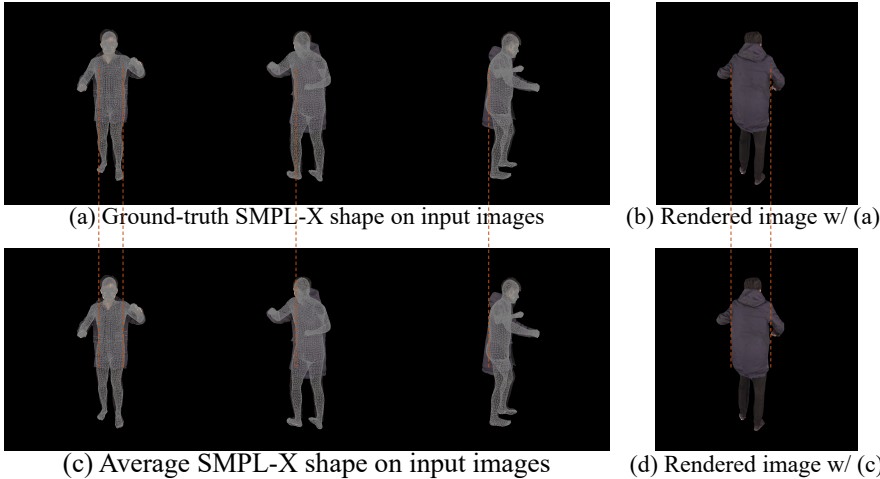

Figure 12: **Robustness to SMPL-X shape accuracy.** We use the ground-truth SMPL-X shape and the average shape as initialization of the canonical mesh. Although the average shape is smaller than the ground-truth shape, our method still captures the correct shape.

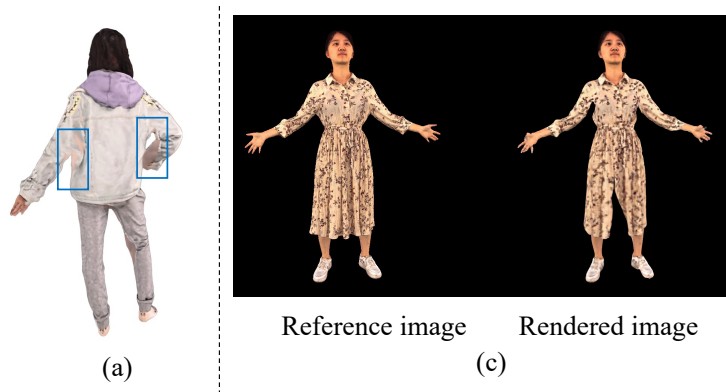

Figure 13: **Examples of failure cases.**

## F  LIMITATIONS

We present three types of failure cases in our method and discuss the possible next steps to resolve the issues.

**Failure in hallucination large regions.** Without an explicit hallucination module, our method is unable to inpaint large invisible regions in source images, as is shown in Fig. 13(a). A possible solution is to render the invisible parts and update our canonical representation using priors from image inpainting models.

**Wrong underlying topology.** Our coupled-multi-resolution Gaussians-on-Mesh representation associates the Gaussians with the underlying mesh. Analogously to the original Gaussians-on-Mesh representation, since the underlying mesh is deformed from human parametric models such as SMPL and SMPL-X, it cannot change vertex connectivities to fit the topology of clothes such as dresses and coats. Although the wrong topology will not affect the rendering, it is a future direction to correct the underlying mesh for use in other downstream tasks.

**Failures for unseen clothing types.** We observe failures for unseen clothing such as dresses, as shown in Fig. 13(b). As a generalizable method, a more comprehensive training set containing different clothings and more diverse subjects is needed. We leave it for future work.

