# OpenReview forum: "LIFe-GoM: Generalizable Human Rendering with Learned Iterative Feedback Over Multi-Resolution Gaussians-on-Mesh"
_ICLR.cc/2025/Conference — ICLR 2025 Poster_

### Official Review · Reviewer_EVBW · 2024-10-27

**Soundness:** 3
**Presentation:** 3
**Contribution:** 3
**Rating:** 8
**Confidence:** 4

**Summary:**

The paper proposes a generalizable human rendering approach using a learned iterative feedback update mechanism. The method leverages a coupled multi-resolution Gaussian-on-mesh representation and demonstrates state-of-the-art results on benchmark datasets such as THuman 2.0 and AIST++.

**Strengths:**

1.	The proposed iterative update net is a novel idea and is clearly motivated.
2.	The writing is clear and easy to follow, with a well-structured introduction and related work section.
3.	Extensive experimentation on benchmark datasets shows clear performance improvements over previous methods (Table 1, Table 2, Table 3).
4.	The authors provide a comprehensive ablation study, which helps justify the effectiveness of iterative step choice and subdivision.

**Weaknesses:**

1. In the related works section, the authors categorize human rendering into "per-scene optimized" and "generalizable" approaches. However, there is no discussion on "large reconstruction model-based" human rendering approaches [1,2]. It would be valuable to compare the proposed method’s performance (in terms of inference time, training time, resolution and PSNR) with such large reconstruction models.

2. The proposed method employs a multi-resolution Gaussian-on-mesh representation, but a direct comparison with traditional mesh-based human representations [3, 4, 5] is missing. It would strengthen the paper if the authors could discuss the advantages or trade-offs between these representations ((in terms of rendering quality, animation flexibility, FPS).

3. The method is built on SMPL/SMPLX models, which might limit its effectiveness in rendering loose clothing, such as skirts or dresses (see  examples in https://zhaofuq.github.io/NeuralAM/). Including examples of such scenarios would enhance the evaluation of the method’s generalizability.

[1] Weng, Zhenzhen, et al. "Single-view 3d human digitalization with large reconstruction models." arXiv preprint arXiv:2401.12175 (2024).

[2] Chen, Jinnan, et al. "Generalizable Human Gaussians from Single-View Image." arXiv preprint arXiv:2406.06050 (2024).

[3] Liao, Tingting, et al. "Tada! text to animatable digital avatars." 2024 International Conference on 3D Vision (3DV). IEEE, 2024.

[4] Zhang, Xuanmeng, et al. "Getavatar: Generative textured meshes for animatable human avatars." Proceedings of the IEEE/CVF International Conference on Computer Vision. 2023.

[5] Liao, Tingting, et al. "High-fidelity clothed avatar reconstruction from a single image." Proceedings of the IEEE/CVF Conference on Computer Vision and Pattern Recognition. 2023.

**Questions:**

While the paper demonstrates promising results, I am curious about the method’s generalizability across diverse scenarios. In particular, on the project page, it would be insightful to evaluate the animation quality of cross-domain examples (e.g., drastically different body types, dynamic poses, or motion sequences). Addressing these aspects would provide a clearer understanding of the robustness and adaptability of the proposed approach

---

> ### Author Response · Authors · 2024-11-19
>
> Thanks for your time and feedback. We appreciate the comments regarding the novelty and effectiveness of our approach. We answer questions next:
>
> ***QD1. In the related works section, the authors categorize human rendering into "per-scene optimized" and "generalizable" approaches. However, there is no discussion on "large reconstruction model-based" human rendering approaches [1,2]. It would be valuable to compare the proposed method’s performance (in terms of inference time, training time, resolution and PSNR) with such large reconstruction models.***
> > In the original submission, we categorize the large reconstruction model-based methods for humans into *diffusion-based methods* in the related works (Line 129) and cite [1]. We thank the reviewer's suggestion and revise the term to *large-reconstruction models* to include a broader set of related works in this direction and cite more works (e.g. [2]). Such approaches usually use a two-step pipeline: 1) they first sample multiview images from diffusion models, and 2) they reconstruct and render the human utilizing the generated multiview images.
> **Discussion:** Note that the task differs from the sparse-view generalizable approach including ours. Large reconstruction model-based methods operate on a single input image, i.e., the key is to hallucinate unseen regions. In contrast, our method focuses on a sparse input setting, and the focus is to align the sparse inputs and reconstruct a canonical T-pose representation. As mentioned in our limitation section, our method cannot hallucinate large unseen regions. Therefore it is only useful to use our method in step 2) of large reconstruction model-based methods. **Comparison:** Unfortunately, since neither [1] nor [2] has released their codes yet, we cannot conduct a fair comparison in PSNR. We alternatively contrast inference speed and training speed based on information provided in the paper:
>
> | Method | Training time and GPU | Reconstruction time and GPU | Rendering time and GPU | resolution  |
> |--------|-----------------------|-----------------------------|------------------------|-------------|
> | [1]    | 7 days on 16xA100     | 0.7s on A6000               | 1.3s on A6000          | 256 x 256   |
> | [2]    | 20h on 4xA6000        | -                           | -                      | 512 x 512   |
> | Ours   | 4 days on 1xA100      | 0.9s on A100                | 10.5ms on A100         | 1024 x 1024 |
>
>
> > In terms of inference speed, [1] mentions that it takes 0.7s for reconstruction from multiview images and 1.3s for rendering at a resolution of $256\times256$ on one NVIDIA A6000 GPU. Our approach uses a higher resolution, i.e., $1024\times1024$ for all stages. Meanwhile, our reconstruction needs 0.9s and the rendering runs in 10.52ms on an NVIDIA A100 GPU. We assume the reconstruction speed of our approach and [1] is close while our rendering speed is faster. For training, [1] trains the coarse single-view model and multiview reconstruction model with 16 A100 GPUs for 7 days, while our model is trained for 4 days on a single A100 GPU.
> > [2] does not report the reconstruction speed and inference speed. For training, it takes 20h on 4 NVIDIA A6000 GPUs while ours takes 4 days on a single A100 GPU. Since we use different resolutions and different GPU types, a fair comparison in terms of training speed is challenging.
>
> [1] Weng, Zhenzhen, et al. "Single-view 3d human digitalization with large reconstruction models." arXiv preprint arXiv:2401.12175 (2024).
>
> [2] Chen, Jinnan, et al. "Generalizable Human Gaussians from Single-View Image." arXiv preprint arXiv:2406.06050 (2024).

---

> > ### Author Response · Authors · 2024-11-19
> > **(2/2)**
> >
> > ***QD2. The proposed method employs a multi-resolution Gaussian-on-mesh representation, but a direct comparison with traditional mesh-based human representations [3, 4, 5] is missing. It would strengthen the paper if the authors could discuss the advantages or trade-offs between these representations ((in terms of rendering quality, animation flexibility, FPS).***
> > > *Thank you. We have added the discussion in revised Appendix E*. The key difference between a mesh representation and Gaussians-on-Mesh is the rendering: Gaussians-on-Mesh leverages the flexibility of Gaussian Splatting, enabling more photorealistic rendering than textured meshes. Their training supervision also differs. Due to the Gaussians-on-Mesh representation, we train our method solely via a photometric loss. However, a mesh is hard to optimize solely via photometric losses, as shown in GoMAvatar [1]. [2] and [3] supervise the rendered 2D normal maps and [4] provides supervision for the 3D normals on the surface. Therefore, mesh-based methods commonly require access to the geometry. In terms of rendering speed and animation flexibility, Gaussians-on-Mesh and mesh representations are both easy to animate and render.
> >
> > [1] Wen, et al. “GoMAvatar: Efficient Animatable Human Modeling from Monocular Video Using Gaussians-on-Mesh”. CVPR 2024
> >
> > [2] Liao, Tingting, et al. "Tada! text to animatable digital avatars." 3DV 2024.
> >
> > [3] Zhang, Xuanmeng, et al. "Getavatar: Generative textured meshes for animatable human avatars." ICCV 2023.
> >
> > [4] Liao, Tingting, et al. "High-fidelity clothed avatar reconstruction from a single image." CVPR 2023.
> >
> > ***QD3. The method is built on SMPL/SMPLX models, which might limit its effectiveness in rendering loose clothing, such as skirts or dresses. Including examples of such scenarios would enhance the evaluation of the method’s generalizability.***
> > > SMPL or SMPL-X shape priors only serve as an initialization in our method. As shown in Eq. (9), our method learns to deform the mesh underlying the Gaussians-on-Mesh representation. Results corroborate this ability. As shown in the first and the third example of the “Freeview Rendering and Comparison to GHG” section of the supplementary webpage, our representation can model clothing which differs from SMPL-X shapes used for initialization, e.g., long coats.
> > > However, we also observe that this isn’t always robust, e.g., for dresses, and mention this in the revised limitations section. Importantly, we think this issue is not caused by SMPL/SMPL-X initialization, as our approach deforms the mesh (e.g., long coats). Instead, we think the issue is due to limited training data: we only trained on 426 subjects, among them __no__ subjects wearing dresses.
> >
> > ***QD4. While the paper demonstrates promising results, I am curious about the method’s generalizability across diverse scenarios. In particular, on the project page, it would be insightful to evaluate the animation quality of cross-domain examples (e.g., drastically different body types, dynamic poses, or motion sequences). Addressing these aspects would provide a clearer understanding of the robustness and adaptability of the proposed approach.***
> > > Thanks for your suggestion. We included more examples in the “Novel pose synthesis” section in the revised supplementary webpage.

---

> > > ### Author Response · Authors · 2024-11-24
> > >
> > > Thank you for providing valuable feedback. As the discussion period is nearing its end, we wanted to check if there are any points that remain unclear or require further clarification from our side. We greatly appreciate your insights and look forward to your response.

---

> > > ### Comment · Reviewer_EVBW · 2024-11-26
> > >
> > > Thanks the authors for their detailed rebuttal. Most of my concerns have been well addressed, and I would like to raise my score to 8.

---

### Official Review · Reviewer_3FuA · 2024-11-02

**Soundness:** 3
**Presentation:** 4
**Contribution:** 3
**Rating:** 6
**Confidence:** 4

**Summary:**

This paper presents a method for generating high-quality, animatable 3D human avatars from sparse images. It proposes a multi-resolution Gaussians-on-Mesh representation, where low-resolution mesh is used for efficient geometry and high-resolution Gaussians for high-quality avatars. This paper also introduces an iterative feedback update mechanism to refine the model.

**Strengths:**

1. The method achieves fast 3D human avatar reconstruction from sparse inputs, making it suitable for real-time applications.
2. The multi-resolution Gaussians-on-Mesh representation balances efficiency and rendering quality.

**Weaknesses:**

1. The experimental comparisons are incomplete without recent works[1,2,3] that report better metrics. A comparative analysis or discussion of technical differences is needed.
2. The cross-domain generalization shows concerning facial artifacts despite the proposed iterative feedback. What causes these failures? If it's a data bias issue, why showcase this case? Consider discussing limitations and potential solutions (e.g., face-aware regularization).
3. The ablation study focuses only on hyperparameters rather than analyzing the effectiveness of core components. Need evaluation of:
Iterative feedback mechanism; Coupled-multi-resolution representation; Module-wise contribution analysis.


[1] 3DGS-Avatar: Animatable Avatars via Deformable 3D Gaussian Splatting, CVPR2024
[2] GoMAvatar: Efficient Animatable Human Modeling from Monocular Video Using Gaussians-on-Mesh, CVPR2024
[3] HUGS: Human Gaussian Splats, CVPR2024

**Questions:**

Please refer to my weakness part. I also have some questions about the videos supplied.
1. Why not show all the input images in Cross-domain generalization part? Without all input frames, it is hard to evaluate the performance.
2. Please make the input number of images clear in Novel pose synthesis part. The novel pose performance is impressive if it is achieved by a single reference image.

---

> ### Author Response · Authors · 2024-11-19
>
> Thanks for your time and feedback. We appreciate the comments regarding the efficiency and the rendering quality of our approach. We answer questions next:
>
> ***QC1. The experimental comparisons are incomplete without recent works[1,2,3] that report better metrics. A comparative analysis or discussion of technical differences is needed.***
> > Thanks for the suggestion! We compare our method to GoMAvatar and 3DGS-Avatar in Table 5 and Table 6 of the revised appendix. The comparisons are as follows: On THuman2.0, taking three source images as input, we obtain
>
> | Method              | PSNR | LPIPS* | FID |
> | :---------------- | ----: | ----: | ----: |
> | GoMAvatar | 23.05 | 133.98 | 87.51 |
> | 3DGS-Avatar | 21.25 | 160.48 | 157.21 |
> | Ours | 24.59 | 111.26 | 51.42 |
>
> > On AIST++, taking five source images as input, we obtain
>
> | Method              | PSNR | SSIM | LPIPS* | Reconstruction time |
> | :---------------- | ----: | ----: | ----: | ----: |
> | GoMAvatar | 24.34 | 0.9780 | 25.34 | ~10h |
> | 3DGS-Avatar | 25.14 | 0.9784 | 27.17 | ~2min |
> | Ours | 25.24 | 0.9809 | 22.11 | 589.27ms |
>
> > In both settings, our approach significantly outperforms GoMAvatar and 3DGS-Avatar in both speed and rendering performance.
>
> > To provide more details about the baselines, we adapt GoMAvatar and 3DGS-Avatar to work with SMPL-X priors on THuman2.0. In both settings, we train GoMAvatar for 100K iterations and 3DGS-Avatar for 2K iterations to avoid overfitting.
>
> [1] 3DGS-Avatar: Animatable Avatars via Deformable 3D Gaussian Splatting, CVPR2024
> [2] GoMAvatar: Efficient Animatable Human Modeling from Monocular Video Using Gaussians-on-Mesh, CVPR2024
> [3] HUGS: Human Gaussian Splats, CVPR2024
>
> ***QC2. The cross-domain generalization shows concerning facial artifacts despite the proposed iterative feedback. What causes these failures? If it's a data bias issue, why showcase this case? Consider discussing limitations and potential solutions (e.g., face-aware regularization).***
> > The distortion in facial regions is mainly due to imperfect poses provided by DNA-Rendering. When the poses are accurate (e.g., for XHuman), we do not observe distortions, as shown in the “Cross-domain generalization - XHuman” section of the supplementary website. We also stress-test our method when human poses and masks are predicted via off-the-shelf tools. Results can be found in the “Cross-domain generalization - PeopleSnapshot” and “Cross-domain generalization - ExAvatar” sections of the supplementary webpage. Note, to demonstrate robustness, on PeopleSnapshot, we do __not__ use the poses and masks provided by the dataset but use predictions instead. Even for predicted poses, our approach shows promising cross-domain generalization.
> > We quantitatively study the sensitivity to input pose accuracy, compare to GHG and discuss the potential solutions in Sec. D.3 of the revised appendix. On THuman2.0, we randomly add Gaussian noise of different standard deviations to the SMPL-X poses. We observe that our approach improves upon the GHG baseline for all noise levels, though all methods are sensitive to the accuracy of input poses:
>
> | Noise  | std=0.1 |        |       | std=0.3 |        |       | std=0.5 |        |       |
> |--------|:-------:|:------:|:-----:|:-------:|:------:|:-----:|:-------:|--------|-------|
> | Method | PSNR    | LPIPS* | FID   | PSNR    | LPIPS* | FID   | PSNR    | LPIPS* | FID   |
> | GHG    | 21.25   | 136.87 | 62.03 | 19.66   | 149.73 | 64.15 | 18.53   | 163.48 | 68.57 |
> | Ours   | 23.96   | 113.80 | 53.15 | 22.02   | 123.15 | 57.22 | 20.43   | 134.86 | 62.84 |
>
> > Additionally, we quantitatively compare our method to the GHG baseline on the XHuman dataset and add those results in Table 7 of the revised appendix. Our method achieves a PSNR/LPIPS*/FID of 25.32/99.32/42.90, improving upon baseline GHG’s 23.53/112.91/50.51. The results show the effectiveness of our approach in cross-domain generalization.
>
> | Method              | PSNR | LPIPS*| FID |
> | :---------------- | ----: | ----: | ----: |
> | GHG | 23.52 | 112.91 | 50.51 |
> | Ours | 25.32 | 99.32 | 42.90 |
>
> ***QC3. The ablation study focuses only on hyperparameters rather than analyzing the effectiveness of core components. Need evaluation of: Iterative feedback mechanism; Coupled-multi-resolution representation; Module-wise contribution analysis.***
> > We would like to clarify that Table 3 and Table 4 showcase the effectiveness of the core components rather than hyperparameters only. “T=1” in Table 3 means no iterative feedback and “#subdivision=0” in Table 4 means w/o coupled-multi-resolution representation, i.e., Gaussians use the mesh resolution. Based on the reported results, both iterative feedback and coupled-multi-resolution representation are important.

---

> > ### Author Response · Authors · 2024-11-19
> > **(2/2)**
> >
> > ***QC4. Why not show all the input images in the cross-domain generalization part? Without all input frames, it is hard to evaluate the performance.***
> > > Great suggestion! We revised the supplementary material to include all the input images.
> >
> > ***QC5. Please make the input number of images clear in novel pose synthesis part. The novel pose performance is impressive if it is achieved by a single reference image.***
> > > For the novel pose synthesis part, our method uses 3 input images. We mention this detail in Line 466 of the revised main paper. The model is identical to the one which was used to obtain the results reported in the first section (“3 source views”) of Table 1 in the main paper and the one used to obtain the results reported in the “Freeview Rendering and Comparison to GHG” section of the supplementary webpage.

---

> > > ### Author Response · Authors · 2024-11-24
> > >
> > > Thank you for providing valuable feedback. As the discussion period is nearing its end, we wanted to check if there are any points that remain unclear or require further clarification from our side. We greatly appreciate your insights and look forward to your response.

---

> > > > ### Comment · Reviewer_3FuA · 2024-11-26
> > > > **Thanks for the clarification.**
> > > >
> > > > I really appreciate the efforts have done during the rebuttal. Considering the comprehensive experiments and the novelty, now I think this paper is ready for ICLR.

---

### Official Review · Reviewer_vbGJ · 2024-11-02

**Soundness:** 3
**Presentation:** 3
**Contribution:** 3
**Rating:** 8
**Confidence:** 3

**Summary:**

The paper tackles the problem of fast, sparse-view 3D articulated avatar modeling. Specifically, the paper proposes to improve Gaussian-on-Mesh method by using a dual-resolution mesh: the low-resolution mesh is used for efficient feed-forward computation, while the high-resolution mesh anchors Gaussians to represent fine details. Furthermore, the approach the sparse input images as cues to extract pixel-aligned features to iteratively improve the 3D avatar. After training, the network can reconstruct 3D avatars for a given identity in under 1-2 seconds.

The experimental results show quantitative and qualitative improvement over previous methods.

After reviewing the rebuttal: my concerns are sufficiently addressed by the new quantitative results provided by the authors. Overall, this submission presents an effective method for generalizable 3D human modeling, and shows non-trivial improvement over the prior art.
I therefore raise my score to 8 -- accept, good paper.

**Strengths:**

The presented method has the following strengths:
- The method shows fast, efficient reconstruction, and this is always a very welcome property for 3D avatar modeling.
- The method shows promising results on reconstructing out-of-distribution avatars.
- The method gets rid of the Gaussian split/growing heuristics, which are sometimes annoying to tune/control.
- The qualitative and quantitative results verify the effectiveness of their proposed design choice.

3D avatar modeling, the problem tackled in this paper, is an important topic in vision/graphics due to its wide impact on many fields (entertainment, sports, virtual try-on, etc). Overall, the presented approach makes sense and the results look promising,

**Weaknesses:**

The paper has the following weakness:
- Missing comparisons to GoMAvatar: as mentioned in line 121-123, the presented method is derived from GoMAvatar. It therefore makes sense to compare against GoMAvatar. The iterative-feedback bears a close similarity to [1], which also utilizes pixel/texel align features to improve fidelity. It would be great if the paper could discuss about these relevant works.
- The writing can be further improved: for example, the sentence in line 143-145: ```As a fix, to regularize the Gaussians and to ease the animation, pairing with parametric models like FLAME or SMPL helps``` may not be as straightforward as saying ```Prior work [cite] regularizes the Gaussians and enables animation using parametric models such as FLAME and SMPL```. Also, in line 180-181 ```given as additional input the target camera ...``` seems unnecessary, as line 175-179 already described the input.
- Mesh topology limits the geometry that can be captured, as shown by the hairs in Figure 4 (also pointed out in Appendix C).
- Missing memory profile: perhaps I have missed it, but I did not see discussions related to the memory/GPU vram consumptions for the proposed method. How much VRAM does it take to train/render the avatars? Does it work on commodity-level GPUs?
- Potentially susceptible to body pose misalignment issues: when the body pose is imperfect/does not overlap with the correct body part, the feedback network will extract features from incorrect locations, and thus impact the reconstruction quality.

[1] Drivable Volumetric Avatars using Texel-Aligned Features, SIGGRAPH 2022

**Questions:**

Please address/discuss about the weakness mentioned above if they are not already in the paper.

---

> ### Author Response · Authors · 2024-11-19
>
> Thanks for your time and feedback. We appreciate the comments regarding the effectiveness and efficiency of our approach. We answer questions next:
>
> ***QB1. Missing comparisons to GoMAvatar: as mentioned in line 121-123, the presented method is derived from GoMAvatar. It therefore makes sense to compare against GoMAvatar.***
> > Thanks for the suggestion! We compare our method to GoMAvatar and add those results as Table 5 and Table 6 in the revised appendix. On THuman2.0, taking three source images as input, we obtain the following result:
>
> | Method              | PSNR | LPIPS*| FID |
> | :---------------- | ----: | ----: | ----: |
> | GoMAvatar | 23.05 | 133.98 | 87.51 |
> | Ours | 24.59 | 111.26 | 51.42 |
>
> > On AIST++, taking five source images as input, we obtain:
>
> | Method              | PSNR | SSIM | LPIPS* |
> | :---------------- | ----: | ----: | ----: |
> | GoMAvatar | 24.34 | 0.9780 | 25.34 |
> | Ours | 25.24 | 0.9809 | 22.11 |
>
> > On both datasets, our approach significantly outperforms GoMAvatar. As a scene-specific method, GoMAvatar tends to overfit the training images in the sparse input setting. In contrast, our method does not suffer from overfitting and works well with only 3 or 5 input images.
>
> ***QB2. The iterative-feedback bears a close similarity to [1], which also utilizes pixel/texel align features to improve fidelity. It would be great if the paper could discuss about these relevant works.***
> > __Iterative feedback__ iteratively updates in a feed-forward manner the canonical representation, given the prediction from the last step. The paper mentioned by the reviewer uses encoder-decoder structures to generate a 3D representation (texels) from input images. Note, the encoder-decoder architecture is not repeated several times, i.e., making it different from our iterative feedback. We highlight the differences in Line 132 of the revised related work section.
>
> [1] Drivable Volumetric Avatars using Texel-Aligned Features, SIGGRAPH 2022
>
> ***QB3. The writing can be further improved: for example, the sentence in line 143-145: As a fix, to regularize the Gaussians and to ease the animation, pairing with parametric models like FLAME or SMPL helps may not be as straightforward as saying Prior work [cite] regularizes the Gaussians and enables animation using parametric models such as FLAME and SMPL. Also, in line 180-181 given as additional input the target camera ... seems unnecessary, as line 175-179 already described the input.***
> > Thanks for the suggestions to improve writing! We revised Line 143-145 accordingly. The poses defined in Line 180-181 and Line 175-179 differ. Specifically, in Line 175-179, we describe the inputs for reconstruction (Eq. (2)), including the camera poses $\\{K\_i, E\_i\\}\_{i=1}^N$ and human poses paired with the $N$ source images. To render the canonical 3D representation, we need to specify the camera pose and human pose. Therefore, in Line 180-181, we formally define the desired camera pose $K\_{tg}, E\_{tg}$ and human pose $P\_{tg} $ for rendering, which can be specified by the user. We apologize for the ambiguity in the notation.
>
> ***QB4. Mesh topology limits the geometry that can be captured, as shown by the hairs in Figure 4 (also pointed out in Appendix C).***
> > Note, SMPL or SMPL-X priors only serve as a mesh initialization in our method. As shown in Eq. (9), the method deforms the mesh underlying our Gaussians-on-Mesh representation. Results corroborate this ability. As shown in the first and the third example of the “Freeview Rendering and Comparison to GHG” section of the supplementary webpage, our method can render clothing which has a topology different from the SMPL topology used for initialization, e.g., long coats, though the connectivity of the underlying mesh is wrong. We defer topology updates to  future work. Thanks for the great suggestion!
>
> ***QB5. Missing memory profile: perhaps I have missed it, but I did not see discussions related to the memory/GPU vram consumptions for the proposed method. How much VRAM does it take to train/render the avatars? Does it work on commodity-level GPUs?***
> > Training requires 30842 MiB VRAM. For reconstruction from given images, we subdivide the mesh once more for higher-quality results, and use 30638 MiB VRAM. Importantly, upon reconstruction, the canonical representation and rendering only uses 2804 MiB VRAM, i.e., rendering runs on commodity-level GPUs once the canonical representation is reconstructed.

---

> > ### Author Response · Authors · 2024-11-19
> > **(2/2)**
> >
> > ***QB6. Potentially susceptible to body pose misalignment issues: when the body pose is imperfect/does not overlap with the correct body part, the feedback network will extract features from incorrect locations, and thus impact the reconstruction quality.***
> > > We quantitatively study the sensitivity to input pose accuracy and compare to GHG in Table 8 of the revised appendix. On THuman2.0, we randomly add Gaussian noise of different standard deviations to the SMPL-X poses. We observe that our approach improves upon the GHG baseline for all noise levels, though all methods are sensitive to the accuracy of input poses:
> > >
> > | Noise  | std=0.1 |        |       | std=0.3 |        |       | std=0.5 |        |       |
> > |--------|:-------:|:------:|:-----:|:-------:|:------:|:-----:|:-------:|--------|-------|
> > | Method | PSNR    | LPIPS* | FID   | PSNR    | LPIPS* | FID   | PSNR    | LPIPS* | FID   |
> > | GHG    | 21.25   | 136.87 | 62.03 | 19.66   | 149.73 | 64.15 | 18.53   | 163.48 | 68.57 |
> > | Ours   | 23.96   | 113.80 | 53.15 | 22.02   | 123.15 | 57.22 | 20.43   | 134.86 | 62.84 |

---

> > > ### Comment · Reviewer_vbGJ · 2024-11-23
> > >
> > > Thanks for the detailed replies, I appreciate the effort.
> > > After going through the reviews from fellow reviewers as well as the rebuttal materials, I believe my main concerns are all addressed.
> > >
> > > Overall, this submission presents an effective method for generalizable 3D human modeling, and shows non-trivial improvement both quantitative and qualitatively. I therefore raise the score to 8 -- accept, good paper.

---

> > > > ### Author Response · Authors · 2024-11-24
> > > >
> > > > We are glad to see the concerns are addressed and thank you for raising the score!

---

### Official Review · Reviewer_5YSm · 2024-11-03

**Soundness:** 3
**Presentation:** 3
**Contribution:** 3
**Rating:** 6
**Confidence:** 5

**Summary:**

This paper tackles the challenge of creating an animatable human avatar from just a few images. To make reconstruction faster and more accurate, they introduce an iterative feedback approach that gradually refines the results. For high-quality, efficient rendering, they also develop a multi-resolution Gaussians-on-Mesh technique, which balances detail and speed.

**Strengths:**

- The feedback mechanism in this generalizable method is notably innovative, refining predictions by addressing visible errors from input views, resulting in more accurate and detailed reconstructions within a second.
- Achieving human avatar reconstruction from images in different poses is highly practical, offering greater flexibility than multi-view approaches.
- The paper is well-written, well-structured, and clear.

**Weaknesses:**

- The model does not appear to have been trained on two datasets and can support both tasks simultaneously, limiting its ability to handle mixed input types effectively.
- The method's heavy dependence on SMPL priors poses a risk; if the estimations are inaccurate, especially for loose clothing, it can lead to significant errors
- The method shows significant performance degradation when applied across domains, particularly affecting the facial region, as in the video. Besides, the novel pose representation displays black hands in the video.
- Minor issues: line 156 "estimate" to "estimation", line 234 translation -> scale?

**Questions:**

- In Figure 1, the mesh appears to be a GT-mesh rather than an SMPL model. Is the GT mesh utilized in your approach？
- Given that Gaussian splatting has a strong tendency to overfit, how is pose generalization achieved in your method? Are the displayed novel poses primarily from the dataset, or do they include truly unseen poses?

---

> ### Author Response · Authors · 2024-11-19
>
> Thanks for your time and feedback. We appreciate comments regarding the novelty of our approach and the more accurate reconstruction. We answer questions next:
>
> ***QA1. The model does not appear to have been trained on two datasets and can support both tasks simultaneously, limiting its ability to handle mixed input types effectively.***
> > *We can handle both tasks simultaneously*. Although our model is trained on only one dataset format/task at a time (multiview for THuman2.0 or multipose for AIST++), inference is not limited to the training dataset format/task. In the updated supplementary material, we apply the model trained on THuman2.0 __multiview__ images to inputs having __different poses__: In the last two examples of the “Cross-domain generalization - XHuman” section of the revised website, we use input images exhibiting different poses. In the “Cross-domain generalization - PeopleSnapshot” and “Cross-domain generalization - ExAvatar” sections of the website, we sample three frames from the monocular video as input. Although the input type differs from the training data, our method shows compelling rendering quality.
> > To further improve results, we agree with the reviewer that joint training on all datasets (THuman2.0 __and__ AIST++) is desirable. However, it is challenging to combine THuman2.0 and AIST++ data, as the two datasets use different body priors: THuman2.0 uses the SMPL-X prior while AIST++ uses the SMPL prior. Nonetheless, combining both datasets for training is an interesting next step.
>
> ***QA2. The method's heavy dependence on SMPL priors poses a risk; if the estimations are inaccurate, especially for loose clothing, it can lead to significant errors.***
> > SMPL or SMPL-X shape priors only serve as an initialization in our method. As shown in Eq. (9), our method learns to deform the mesh underlying the Gaussians-on-Mesh representation. Results corroborate this ability. As shown in the first and the third example of the “Freeview Rendering and Comparison to GHG” section of the supplementary website, our representation can model clothing which differs from SMPL-X shapes used for initialization, e.g., long coats.
> > However, we also observe that this isn’t always robust, e.g., for dresses, and mention this in the revised limitations section. Importantly, we think this issue is not caused by SMPL/SMPL-X initialization, as our approach deforms the mesh (e.g., long coats). Instead, we think the issue is due to limited training data: we only trained on 426 subjects, among them __no__ subjects wearing dresses.
>
> ***QA3. The method shows significant performance degradation when applied across domains, particularly affecting the facial region, as in the video.***
> > To study cross-domain generalization, we quantitatively compare our method to the GHG baseline on the XHuman dataset and add those results in Table 7 of the revised appendix. Our method achieves a PSNR/LPIPS*/FID of 25.32/99.32/42.90, improving upon baseline GHG’s 23.53/112.91/50.51.
>
> | Method              | PSNR | LPIPS*| FID |
> | :---------------- | ----: | ----: | ----: |
> | GHG | 23.52 | 112.91 | 50.51 |
> | Ours | 25.32 | 99.32 | 42.90 |
>
> > We further demonstrate cross-domain generalization in the revised supplementary website (see “Cross-domain generalization - XHuman” section in the supplementary website).
>
> > Note, the distortion in facial regions is mainly due to imperfect poses provided by DNA-Rendering. When the poses are accurate (e.g., for XHuman), we do not observe distortions, as shown in the “Cross-domain generalization - XHuman” section of the supplementary website. We also stress-test our method when human poses and masks are predicted from off-the-shelf tools. Results can be found in the “Cross-domain generalization - PeopleSnapshot” and “Cross-domain generalization - ExAvatar” sections of the supplementary webpage. Note, to demonstrate robustness, on PeopleSnapshot, we do __not__ use the poses and masks provided by the dataset but use predictions instead. Even for predicted poses, our approach shows promising cross-domain generalization.
>
> ***QA4. The novel pose representation displays black hands in the video.***
> > The black hands observed for novel pose synthesis are due to occlusions: note that the hand is in the pocket in the reference source images and is therefore entirely invisible. As mentioned in the original limitation section, our method doesn’t hallucinate large unseen regions.
>
> ***QA5. Minor issues: line 156 "estimate" to "estimation", line 234 translation -> scale?***
> > Thanks for pointing this out, we updated the paper accordingly.

---

> > ### Author Response · Authors · 2024-11-19
> > **(2/2)**
> >
> > ***QA6. In Figure 1, the mesh appears to be a GT-mesh rather than an SMPL model. Is the GT mesh utilized in your approach?***
> > > __We do not use a GT mesh__. The mesh in Figure 1 is the final mesh, predicted by our proposed method. We obtain it by initializing the mesh of the Gaussians-on-Mesh representation with the SMPL or SMPL-X __prior__, and subsequently refine it during inference by predicting vertex offsets (see Eq. (9)).
> >
> > ***QA7. Given that Gaussian splatting has a strong tendency to overfit, how is pose generalization achieved in your method? Are the displayed novel poses primarily from the dataset, or do they include truly unseen poses?***
> > > __Our method generalizes to truly unseen poses, which is what we show in our paper__. The novel poses we use in the qualitative results (Fig. 6 and “Novel pose synthesis” section on the website) are unseen poses captured from the BEDLAM dataset. In the reconstruction phase, our method reconstructs the subject in the canonical T-pose instead of the pose of a/the source images. The canonical representation can be animated arbitrarily via linear blend skinning (LBS) following Eq. (5).

---

> > > ### Author Response · Authors · 2024-11-24
> > >
> > > Thank you for providing valuable feedback. As the discussion period is nearing its end, we wanted to check if there are any points that remain unclear or require further clarification from our side.
> > > We greatly appreciate your insights and look forward to your response.

---

> > > > ### Comment · Reviewer_5YSm · 2024-11-25
> > > > **FINAL score**
> > > >
> > > > hello, I insist on my score for baseline acceptance.
> > > > I suggest the author provide more ablation studies about erroneous SMPL  input since this concern is not addressed well.

---

> > > > > ### Author Response · Authors · 2024-11-26
> > > > >
> > > > > Thanks for the suggestion! Our method is generally robust to the SMPL-X shape accuracy. We ablate the sensitivity to SMPL-X shapes quantitatively and qualitatively. For this, we initialize the canonical mesh with the average SMPL-X shape by setting the beta parameter to all zeros for all subjects. Neither ground-truth nor predicted SMPL-X shapes are used for any subject. We obtain the following results:
> > > > >
> > > > > |                      | PSNR  | LPIPS* | FID   |
> > > > > |----------------------|-------|--------|-------|
> > > > > | GHG w/ SMPL shapes   | 21.90 | 133.41 | 61.67 |
> > > > > | Ours w/o SMPL shapes | 24.19 | 113.07 | 51.67 |
> > > > > | Ours w/ SMPL shapes  | 24.59 | 111.26 | 51.42 |
> > > > >
> > > > > Our method attains PSNR/LPIPS*/FID of 24.19/113.07/51.67 w/o SMPL-X shapes as input. Compared to 24.59/111.26/51.42 with SMPL-X shapes as input, we only observe a small drop. Even **without** SMPL-X shapes, ours still significantly outperforms GHG's 21.90/133.41/61.67 **with** SMPL-X shapes as input. We add qualitative results in Fig. 12 in the revised appendix. We also append the ablation study to Appendix D.4. We hope the results answer your question. Please let us know if there are additional experiments to strengthen the conclusion!

---

### Author Response · Authors · 2024-11-19
**Reply to all reviewers**

We thank all reviewers for their detailed comments, and for finding our work innovative (Reviewer 5YSm, EVBW), effective (Reviewer vbGJ, EVBW) and practical (Reviewer 5YSm). We updated the paper and supplementary material accordingly and highlighted the new content in blue. Additions include 1) the comparison to scene-specific methods in Appendix D.1; 2) quantitative and qualitative results in cross-domain generalization in Appendix D.2 and in the “Cross-domain generalization” section of the revised website; 3) analysis of input pose sensitivity in Appendix D.3; 4) more novel pose synthesis results in the “Novel pose synthesis” section of the revised website; and 5) related works on mesh-based representations in Appendix E. We address questions individually.

---

### Comment · Area_Chair_9pWb · 2024-11-25
**Please read the rebuttal and reply**

Dear Reviewers,

Thanks again for serving for ICLR, the discussion period between authors and reviewers is approaching (November 27 at 11:59pm AoE), please read the rebuttal and ask questions if you have any. Your timely response is important and highly appreciated.

Thanks, AC

---

### Meta-Review · Area_Chair_9pWb · 2024-12-23

**Metareview:**

This paper proposes a method to reconstruct animatable 3D avatars from sparse input images. Specifically, it develops a novel Multi-resolution Gaussians-on-Mesh representation, together with an iterative feed-back mechanism. This paper resolves an important task with a novel and effective method.

During rebuttal, reviewers raised questions including the dependency on SMPL pose estimation, inferior performance for cross-domains data, missing comparisons, etc. After rebuttal, most concerns have been addressed and all reviewers agree to accept this paper.

**Additional Comments On Reviewer Discussion:**

During rebuttal, reviewers raised the following issues:
- the dependency on SMPL pose estimation (Reviewer 5YSm, vbGJ, EVBW)
- performance degradation when applied across domains (Reviewer 5YSm, 3FuA)
- Missing comparisons with existing works (Reviewer vbGJ, 3FuA, EVBW)
- Missing memory profile (Reviewer vbGJ)
- Missing ablation studies (Review 3FuA)

After rebuttal, most concerns have been addressed and all reviewers agree to accept this paper.

---

### Decision · Program_Chairs · 2025-01-22

Accept (Poster)